# Amines in Boreal Forest Air at SMEAR II Station in Finland

Marja Hemmilä[1], Heidi Hellén[1], Aki Virkkula[1,2], Ulla Makkonen[1], Arnaud P. Praplan[1], Jenni Kontkanen[2], Lauri Ahonen[2], Markku Kulmala[2], Hannele Hakola[1]

[1] Finnish Meteorological Institute, P.O. Box 503, FI-00101 Helsinki, Finland
[2] Institute for Atmospheric and Earth System Research/Physics, Faculty of Science, University of Helsinki, P.O. Box 64, 00014 Helsinki, Finland

*Correspondence to:* Marja Hemmilä (marja.hemmila@fmi.fi)

**Abstract.** We measured amines in boreal forest air in Finland both in gas and particle phase with 1-hour time resolution using an online ion chromatograph (instrument for Measuring AeRosols and Gases in Ambient Air, MARGA) connected to an electrospray ionization quadrupole mass spectrometer (MS). The developed MARGA-MS method was able to separate and detect 7 different amines: monomethylamine (MMA), dimethylamine (DMA), trimethylamine (TMA), ethylamine (EA), diethylamine (DEA), propylamine (PA) and butylamine (BA). The detection limits of the method for amines were low (0.2–3.1 ng m$^{-3}$), the accuracy of IC-MS analysis was 11–37% and the precision 10–15%. The proper measurements in the boreal forest covered about 8 weeks between March 2015 and December 2015. The amines were found to be an inhomogeneous group of compounds, showing different seasonal and diurnal variability. Total MMA (MMA(tot)) peaked together with the sum of ammonia and ammonium ion already in March. In March monthly means for MMA were <2.4 ng m$^{-3}$ and 6.8±9.1 ng m$^{-3}$ in gas and aerosol phase, respectively, and for NH$_3$ and NH$_4^+$, 52±16 ng m$^{-3}$ and 425±371 ng m$^{-3}$, respectively. Monthly medians in March for MMA(tot), NH$_3$ and NH$_4^+$, were <2.4 ng m$^{-3}$, 19 ng m$^{-3}$ and 90 ng m$^{-3}$ respectively. DMA(tot) and TMA(tot) had summer maxima indicating biogenic sources. We observed diurnal variation for DMA(tot) but not for TMA(tot). The highest concentrations of these compounds were measured in July. Then monthly means for DMA were <3.1 ng m$^{-3}$ and 8.4±3.1 ng m$^{-3}$ in gas and aerosol phase, respectively, and for TMA 0.4±0.1 ng m$^{-3}$ and 1.8±0.5 ng m$^{-3}$. Monthly medians in July for DMA were <DL and 4.9 ng m$^{-3}$ in gas and aerosol phase, respectively, and for TMA 0.4 ng m$^{-3}$ and 1.4 ng m$^{-3}$. When relative humidity of air was >90%, gas phase DMA correlated well with 1.1–2 nm particle number concentration (R$^2$=0.63) suggesting that it participates in atmospheric clustering. EA concentrations were low all the time. Its July means were <0.36 ng m$^{-3}$ and 0.4±0.4 ng m$^{-3}$ in gas and aerosol phase respectively, but individual concentration data correlated well with monoterpene concentrations in July. Monthly means of PA and BA were all the time below detection limits.

## 1. Introduction

In atmospheric chemistry and secondary-aerosol production bases are crucial since they can neutralize acids and therefore accelerate several processes, like e.g. subsequent growth of newly born aerosol particles. Furthermore bases are significant since they diminish acidification. Amines are gaseous bases, whose general formula is $RNH_2$, $R_2NH$ or $R_3N$. Due to their effective participation in neutralization it is hard to detect their real atmospheric concentrations. Globally, the main known anthropogenic amine emissions are from animal husbandry, industry and compost processes, and the natural sources of amines are assumed to be ocean, biomass burning, vegetation and soil (Ge et al., 2011). It has been shown that amines affect hydroxyl radical (OH) reactivity and therefore all atmospheric chemistry (Hellén et al. 2014, Kieloaho et al. 2013).

Models based on quantum chemistry data have shown that amines could participate in new particle formation (NPF) with sulfuric acid even at very low mixing ratios (Kurtén et al. 2008, Paasonen et al. 2012), and also experiments in laboratory have proved formation of aminium salts when amines react with nitric or sulphuric acid (Murphy et al. 2007). In addition the recent experiments at the CLOUD chamber shows that even at minute concentrations of dimethylamine (DMA) new particles with sulphuric acid are produced (Almeida et al. 2013, Kürten et al. 2016). Atmospheric aerosols affect the climate, because they can act as cloud condensation nuclei (IPCC 2014). They also scatter and absorb solar radiation.

Ambient concentrations of gas-phase amines have been measured earlier using different methods: samples have been collected in phosphoric-acid-impregnated fiberglass filters (Kieloaho et al. 2013), to solid phase micro extraction fiber (SPME, Parshintsev et al. 2015), and to ion-exchange resin (Dawson et al. 2014) and they have also been percolated through an acidic solution (Akyüz 2007). Samples have been analyzed later in the laboratory with various chromatographic techniques, such as gas chromatography coupled to mass spectrometry (GC-MS) (Akyüz 2007, Parshintsev et al., 2015), ion chromatography (IC) (Dawson et al. 2014) and high performance liquid chromatography coupled to mass spectrometry (HPLC-MS) (Kieloaho et al. 2013). The above mentioned techniques have various shortcomings: quantitation based on collection onto fibers is problematic, collecting in filters requires long sampling times (usually several days), and percolating in acidic solutions requires intensive sample pre-treatment. Dawson et al. (2014) used weak cation exchange resin as a substrate for collection of gas-phase ammonia and amines. The method minimizes sample losses on walls during sampling and has quite short sampling times (less than an hour), but the detection limits remain too high for the boreal forest environment.

In addition novel in-situ methods for measuring ambient air gas-phase amines have been developed, usually based on mass-spectrometric detection: chemical ionization mass spectrometry (CIMS), (Sellegri et al. 2005, You et al. 2014), ambient pressure proton transfer mass spectrometry (AmPMS) (Hanson et al. 2011, Freshour et al. 2014), chemical ionization

atmospheric pressure interface time-of-flight mass spectrometry (CI-APi-TOF) (Kulmala et al. 2013, Sipilä et al. 2015, Kürten et al. 2016) and TOF-CIMS (Zheng et al 2015). These in-situ techniques have short time resolution and the limits of detections are small. However, these methods cannot separate amines with same masses (e.g. DMA and EA) and identification of the measured compounds remains uncertain. Chang et al. (2003) used high-efficiency planar diffusion scrubber IC (HEDS-IC) to successfully separate amines with identical masses.

Aerosol phase amines have been sampled onto filters and analyzed later in the laboratory with similar techniques: LC-MS (Ruiz-Jiménez et al. 2012), GC-MS (Huang et al. 2014) and IC (Huang et al. 2014, van Pinxteren et al. 2015). With these methods sampling time was long (24–133 h) and biases may be introduced due to transport and pretreatment of samples. VandenBoer et al. (2011) measured amine concentrations both in gas and particle phase with an ambient ion monitor –IC (AIM-IC). This method had 60-min sampling time and relatively low detection limits (5–9 ng m$^{-3}$). However, it could not separate TMA and DEA from each other. Also because in atmospheric samples ammonia/ammonium can be present in concentrations several orders of magnitude higher than amines in this method they can impede detection of some amines (e.g. MMA and EA).

These methods have been utilized in short campaigns from a couple of days to a couple of weeks. Only Kieloaho et al. (2013) measured for a longer period, but their sampling time was long (24–72 h). Most of the studies discussed previously were made in urban or sub-urban areas, and only a few measurements (Sellegri et al. 2005, Kieloaho et al. 2013, Kulmala et al. 2013 and Sipilä et al. 2015) were made in a boreal forest. In these studies the observed alkylamine concentrations ranged from below detection limit to ~150ppt$_v$, depending on the sampling time and the analysis method used.

Here we present the in-situ method developed for atmospheric amine measurements in this study, using an online ion chromatography, instrument for Measuring AeRosols and Gases in Ambient air, coupled with mass spectrometer (MARGA-MS). The method was used in the boreal forest, where amines are expected to affect secondary aerosol formation even at extremely low concentrations (Kurtén et al. 2008, Paasonen et al. 2012, Almeida et al. 2013). We report seasonal and diurnal variations of amines in boreal forest air and their partitioning between gas and aerosol phase. A time series of diurnal observations and linkages to known boreal biogenic processes is discussed for several amines. Our investigation is the first long-term survey of sources and phase-distribution of amines at the sub-pptv level in a remote boreal forest environment. In this study we use supporting physical measurements to initiate a better understanding of this entire class of compounds relative to what we already know about ammonia.

## 2. Experimental

We measured amine and ammonia concentrations in 2015 from March to May (spring), July to August (summer) and November to December (early winter) with one-hour time resolution. However, due to instrumental problems, good-quality data were captured for a total only about 8 weeks.

### 2.1 Measurement site

Measurements were performed in a Scots pine forest at the SMEAR II station (Station for Measuring Forest Ecosystem-Atmosphere Relations) in Hyytiälä, southern Finland (61°51´N, 24°17´E, 180 m a.s.l., Hari and Kulmala, 2005, Fig. S1). The largest nearby city is Tampere, situated 60 km southwest from the station with approximately 222 000 inhabitants in the city itself (although 364 000 in the wider metropolitan area). The instrument was located in a container about 4 meters outside the forest in a small opening. In addition to pines, also small spruces (Picea abies) grow nearby. The forest was planted about 50 years ago and its current tree height is about 19 m.

### 2.2 Meteorological conditions

Meteorological quantities were obtained from the SmartSmear AVAA portal (Junninen et al. 2009). SmartSmear is the data portal for visualization and download of continuous atmospheric, flux, soil, tree, physiological and water quality measurements at SMEAR research stations of the University of Helsinki. Table S1 shows the meteorological conditions during measurements periods.

### 2.3 Measurement methods

#### 2.3.1. MARGA-MS

We used the instrument for Measuring AeRosols and Gases in Ambient air (MARGA, Metrohm-Applikon, Schiedam, Netherlands) (ten Brink et al. 2007) for sampling and separating amines. MARGA is an online ion chromatograph (IC) connected to a sampling system. In addition, this system was connected to an electrospray ionization (ESI) quadrupole MS (Shimadzu LCMS-2020, Shimadzu Corporation, Kyoto, Japan) to improve sensitivity of amine measurements (see Table S2 for MS settings). The MARGA instrument earlier used for measuring anions and cations in Helsinki and Hyytiälä is described in more detailed in earlier papers (Makkonen et al. 2012 and 2014).

Ambient air was taken through a PM10 cyclone (URG 1032, Teflon coated) and polyethylene tubing (ID 0.5", length ~1 m) with a flow rate of 16.7 l min$^{-1}$. After passing the inlet, sample air entered to a wet rotating denuder (WRD), where the gases diffused into the absorption solution (10 ppm hydrogen peroxide). Particles passed through the WRD and entered the steam jet aerosol collector (SJAC), where they were collected in a supersaturated environment (in 10 ppm hydrogen peroxide). During each hour liquid samples from the WRD and SJAC were collected in the syringes (25 ml), mixed with the internal standard (LiBr and deuterated diethyl-d10-amine) and injected to the cation ion chromatograph. The two sets of syringes worked in tandem, so that when a set of samples was collected, the previous ones were injected. In the cation chromatograph 3.2 mmol l$^{-1}$ oxalic acid (Merck, Darmstadt, Germany) solution was used as an eluent (constant flow 0.7 ml min$^{-1}$). To get the detection limits lower we used a concentration column (Metrosep C PCC 1 VHC/4.0) before the analytical column (Metrosep C4-100/4.0, 100 mm x 4.0 mm i.d., stationary phase silica gel with carboxyl groups, particle size 5 μm). After passing the cation column and the conductivity detector, samples were guided to the ESI-needle of the mass spectrometer without any additional solvent. All solutions used were made with ultrapure water (Milli-Q, resistivity ≥18 MΩ·cm)

Detection limits (DL) for MARGA-MS were calculated from signal-to-noise ratios (3:1) for most of the compounds and they were similar in gas and aerosol phase, because their blank-values were so small (Table 1 in section 3.1). However, DLs for DMA and TMA were calculated from blank-values (3 times standard deviations of blank-values) and the DLs were different for gas and aerosol-phase measurements.

Deuterated diethyl-d$_{10}$-amine (DEA$_{10}$, Sigma-Aldrich: Isotec™; Sigma-Aldrich, St. Louis, MO, USA) was used as an internal standard (ISTD) for all amines. DEA$_{10}$ was used, because it behaved same way in IC-separation but had different mass than studied amines. 50.0μl of DEA$_{10}$ was added to the MARGAs ISTD solution bottle (LiBr). After the ion chromatograph the ISTD mixed with the sample entered the MS-detection. DEA$_{10}$ was used to correct for possible losses to instrumentation and correct changes of MS response. A 3-point external calibration was used for all measured alkyl amines (concentration levels 10, 50 and 300 ng m$^{-3}$). The system was calibrated every two weeks, by stopping the air flow of the MARGA and directing standard solutions to the sample syringe pumps, before analysis by IC-separation and MS-detection. Ammonia (NH$_3$) and ammonium (NH$_4^+$) (the sum of them referred to as NH$_x$) were also measured with MARGA at the same time with the method described in Makkonen et al. (2012 and 2014), except we used oxalic acid solution for eluent. For NH$_x$-measurements only conductivity detector was used and the internal standard was lithium bromide (Acros Organics, New Jersey, USA). Instrumental blank values for MARGA-MS were measured every month or every other month with MARGAs blank-mode: the sample airflow was stopped, and the analysis cycle was running for 6 hours without sampling.

In calculations the values under DLs were taken account as 0.5×DL. In the figures we used a moving average for DMA, because every other measured DMA concentration was a little higher than the in-between one. The system used different

syringes for sample collection every other hour and the reason for differences are expected to be losses or contamination in the syringes. Further causes for these minor differences were not found.

### 2.3.2 Aerosol measurements

To study the role of amines in atmospheric particle formation, particle number concentration measurements were utilized. The particle number size distribution between 3 and 1000 nm was measured with a twin- Differential Mobility Particle Sizer (DMPS) system (Aalto et al. 2001). From these measurements, the particle concentration between 3 and 25 nm ($N_{3-25\ nm}$),

referred to as the nucleation mode, and the total particle concentration between 3 and 1000 nm ($N_{tot}$) were obtained. In addition, the concentrations of sub-3 nm particles were measured with an Airmodus Particle Size Magnifier (PSM A11; Vanhanen et al. 2011). The PSM is a mixing-type condensation particle counter, in which particles are first grown to 90 nm size by condensation of diethylene glycol, after which butanol is used to grow them to detectable sizes. The cut-off size of the PSM can be changed by altering the mixing ratios of saturated and sample flows, which allows the measurement of

particle size distribution in the sub-3 nm size range. In this study, the particle concentration obtained for the size range between 1.1 and 2.0 nm ($N_{1.1-2nm}$) was used. In addition, the particle concentration between 2 and 3 nm ($N_{2-3nm}$), was obtained by subtracting the total particle concentration measured with the highest cut-off size of the PSM from the total particle concentration measured with the DMPS. For more discussion about the particle concentration measurements and their uncertainties, see Kontkanen et al. (2017) who have published the data set used in this study.

### 2.4 Regression calculations

Simple linear regressions were calculated to find whether meteorological conditions affect amine concentrations. The statistical significance of the slope of the linear regression of the amine concentration y vs. the ambient condition x, i.e. $y = \beta_1 x + \beta_0$ was estimated. The null hypothesis, which means that the slope $\beta_1$ is not dependent on the ambient condition x (i.e.,

$\beta_1 = 0$), was examined using test statistics given by the estimate of the slope divided by its standard error ($t = \beta_1/s.e.$). The test statistics were compared with the Student's t distribution on n - 2 (sample size minus the number of regression coefficients) degrees of freedom. The analysis yields also the p value of the slope. The lower the p-value is, the stronger the evidence against the null hypothesis is.  The statistical significance of the slope can be interpreted so that if $p > 0.1$ there is no evidence against the null hypothesis, and p-values in the ranges 0.05-0.1, 0.01-0.05, and $< 0.01$ suggest respectively a

weak, moderate and strong evidence against the null hypothesis in favor of the alternative. The regressions were calculated for amine concentrations vs. air temperature, relative humidity, wind speed, soil temperature and soil humidity.

## 3. Results

### 3.1 Characterization of MARGA-MS

An on-line method for sampling, separating and detecting amines from the ambient air both in the gas and aerosol phase has been developed. With MARGA-MS we studied 7 different amines: monomethylamine (MMA), dimethylamine (DMA), trimethylamine (TMA), ethylamine (EA), diethylamine (DEA), propylamine (PA) and butylamine (BA), see Figure S2 for the chromatogram. The time resolution of measurements was one hour, and as can be seen in Table 1, the detection limits were low, and precision (10–15%) and accuracy (11–37%) for the analytical method of MARGA-MS were moderately good. In addition to improved DLs, MS detection after MARGA also solved the problem with co-elution of amines with different molecular masses and inorganic cations (e.g. K+, Mg2+). Verriele et al. (2012) developed also an IC-MS method for amines with offline sampling with midget impingers. They also noticed that adding MS detection after a conductivity detector overcomes the co-eluting problem of IC separation. They had a 4-step gradient elution in their method, and suppression before the conductivity detector. We wanted to keep our method as simple as possible to make it easy to use in the field, and isocratic elution without suppression was good in that purpose.

The whole analysis was conducted in the field, so the method had no biases from sample transportation. However, the drawback in the analysis was that DEA and BA, which have the same molecular masses, did not separate completely. From a technical point of view one of the drawbacks of the MARGA-MS was that the system was quite vulnerable. We lost many measuring days because some part of the system was broken. The MARGA side also needed ~40 l solutions (e.g. eluents, absorbation solution for sampling, and internal standard solution) that needed to be changed weekly. The ESI-chamber of the MS needed to be cleaned weekly, because oxalic acid was crystallizing into it. Despite the drawbacks, to our knowledge with the MARGA-MS method we achieved the largest data set of amine concentrations available at the moment.

Table 1. Detection limits (DL) of different amines, ammonia and ammonium. Conversions from (ng m$^{-3}$) to ppt$_v$ has been made using conversion factor ppt$_v$ = c(ng m$^{-3}$) : (0.0409×(MW)) by Finlayson-Pitts (2000), with MW the molar mass of the amine, ammonia or ammonium.. The precision for IC-MS analysis was defined by calculating standard deviations of liquid 200 ng m$^{-3}$ standard measured 6 times in a row. In the data series there were both gas and particle side measurements. The accuracy for IC-MS analysis was calculated by subtracting the averages of the data series described earlier from the expected values, dividing those with the expected values and multiplying them by 100%.

| Amine | | DL (ng m$^{-3}$) | DL (ppt$_v$) | Precision (%) | Accuracy (%) |
|---|---|---|---|---|---|
| MMA, | both gas and aerosol | 2.4 | 1.9 | 10 | 24 |
| DMA, (March to August) | gas | 3.1 | 1.7 | 11 | 31 |
| | aerosols | 1.1 | | | |
| (November to December) | gas | 0.37 | 0.20 | | |
| | aerosols | 0.76 | | | |
| TMA, | gas | 0.2 | 0.1 | 14 | 11 |
| | aerosols | 0.5 | | | |
| EA, | both gas and aerosol | 0.36 | 0.19 | 11 | 16 |
| DEA, | both gas and aerosol | 0.24 | 0.08 | 15 | 37 |
| PA, | both gas and aerosol | 0.31 | 0.13 | 11 | 21 |
| BA, | both gas and aerosol | 0.26 | 0.09 | 12 | 14 |
| NH$_3$, | gas | 11.4 | 16.4 | | |
| NH$_4^+$, | aerosol | 2.9 | | | |

### 3.1.1 Particle collection in the denuder of the MARGA

Theoretical calculations of diffusional losses through an annular tube have been derived, e.g., by Winiwarter (1989). The numerical solution of the diffusional losses in an annular denuder presented by Fan et al. (1996) and Baron and Willeke (2001) were applied to calculate the size-dependent penetration in the denuder of the MARGA. The calculation needs as input the diameter of the inner and outer tubes (36.4 mm and 39.9 mm, respectively), the tube length 26.5 cm and the flow rate 16.7 LPM.

The result of the calculation (Fig. 1) shows that 50% of particles smaller than about 6 nm are collected in the denuder and get interpreted as gas-phase compounds. It also shows that essentially all particles larger than about 20 nm get transported through the denuder and finally get interpreted correctly as particles. The cluster-mode particles are smaller than 2 nm and behave primarily like gases and more than ~80% of them do not penetrate the denuder, whereas more than ~85% of particles

larger than 10 nm go through it. The size of nucleation-mode particles is approximately between 2–10 nm, or up to ~25 nm, depending on the definition of the size ranges, and they appear in the atmosphere mainly during NPF events. During these events they could be found both in the denuder and in the steam-jet aerosol collector, but that does not play an essential role because of their small mass even when the number concentration is high. An estimate of the masses involved can be given by assuming that the number concentration in a nucleation mode is $10\ 000\ \text{cm}^{-3}$, its geometric mean diameter $D_g = 4$ nm, and

the geometric standard deviation $\sigma_g = 1.5$. Assuming that the density of particles is $1.5\ \text{g cm}^{-3}$ the mass of that mode is ~1.05 ng m$^{-3}$. The diffusion losses in the denuder result in a growth of the geometric mean of the size distribution and decrease of mass concentration to ~0.69 ng m$^{-3}$ which means that 65 % of the mass gets into the SJAC. The fraction of mass penetrating to the SJAC grows with a growing modal diameter so that for a single-mode distribution of $D_g = 10$ nm and $\sigma_g = 1.5$ the penetrated mass fraction is 95%.


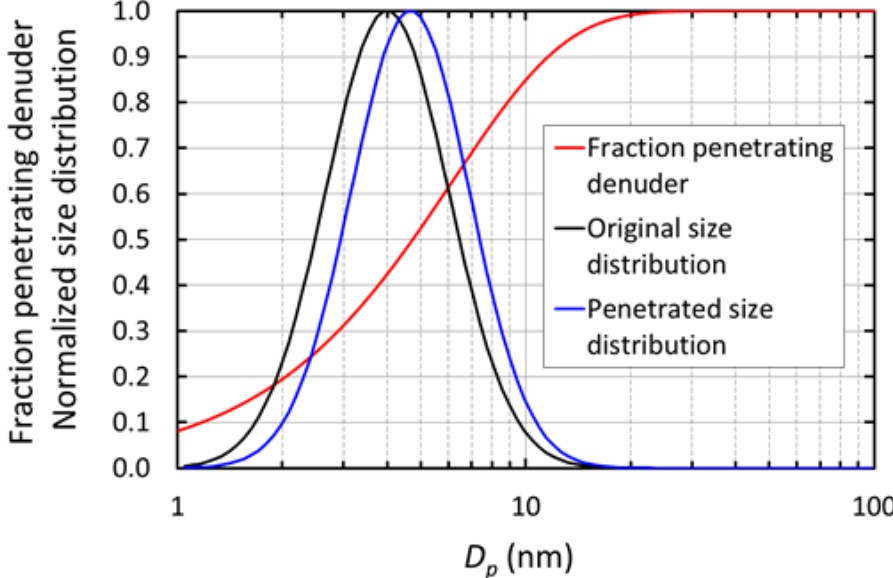

Figure 1. Size-dependent particle penetration probability in the annular denuder of the MARGA and a normalized number size distribution (dN/dlogD$_p$, $D_g = 4$ nm, $\sigma_g = 1.5$) of a nucleation mode before and after penetrating the denuder.

## 3.2 Variability of the concentrations

Figure S4 shows the monthly means and medians of total amine concentrations (tot, sum of gas and aerosol phases) and Figure 2 shows the box and whisker plots to describe the distribution of the measured concentrations. Total amine concentrations were used because we wanted to study how amine sources and partitioning between aerosol (a) and gas phase (g) depend on environmental quantities. Even though the average ratios (gas/(gas+aerosol)) for values above DL in Table 2 are close to 0.5, amines were still mainly in the aerosol phase (Table 2 and S2), which is shown by the more data points >DL

in the aerosol phase. Table S3 shows the number of data points in each month, as well as the mean and median values of concentrations of different amines, ammonia and ammonium. It can be seen, that most concentrations were below DL especially in the gas phase, so we can conclude that concentrations of amines in the boreal forest are low compared to for example ammonia or monoterpene concentrations (Hakola et al. 2012). In Table 3, concentrations in other studies are compared to our findings. Different seasonal patterns were found for different amines and they are described below.


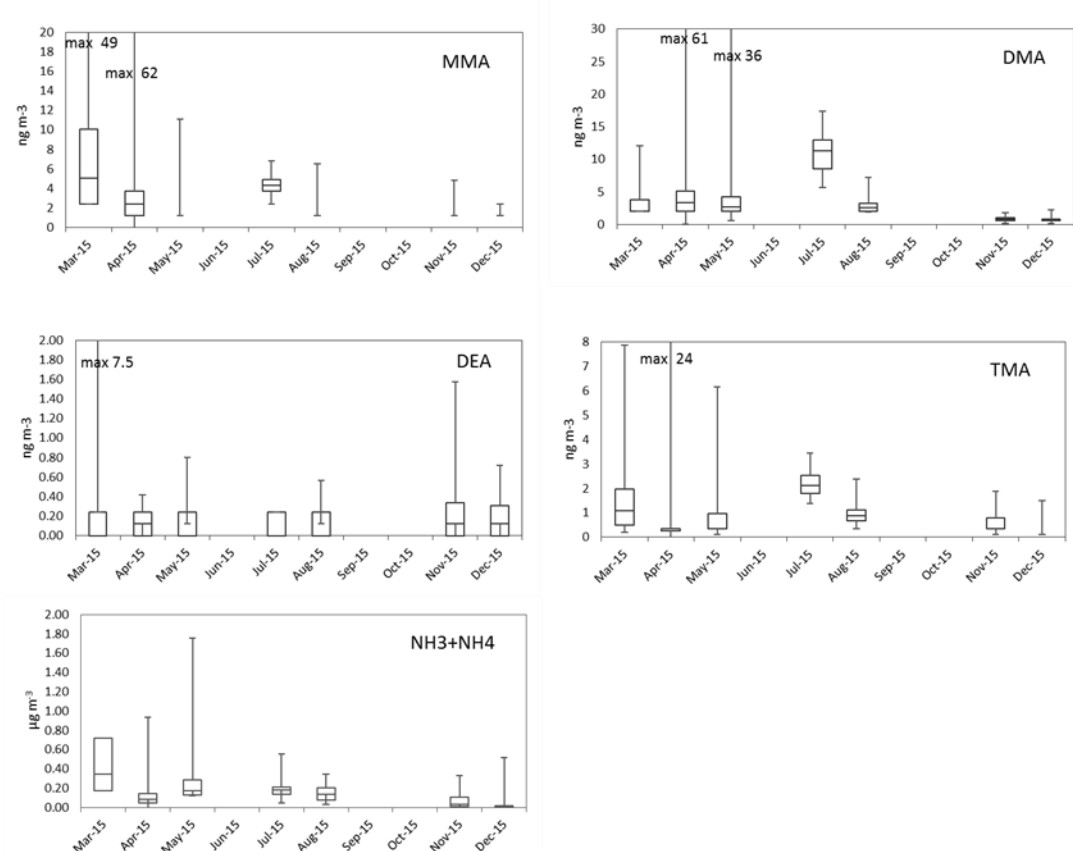

Figure 2: Monthly box and whisker plots of the most abundant amines (tot) and summed up ammonia and ammonium. The boxes represent the second and third quartiles and the lines in the boxes the median values. The whiskers show the highest and the lowest observations.

Table 2. Ratio of gas and aerosol phases. N(g)=number of gas phase data above detection limit (DL), N(a)=number of aerosol phase data above DL and N=number of data above DL at the same time both in the gas and aerosol phases, ratio=gas/(gas+aerosol) (when both values were above the DL).

|  | MMA | DMA | TMA | EA | DEA | PA | BA | NH$_x$ |
|---|---|---|---|---|---|---|---|---|
| N(g) | 29 | 116 | 308 | 62 | 86 | 20 | 38 | 285 |
| N(a) | 183 | 550 | 391 | 82 | 29 | 35 | 26 | 844 |
| N | 9 | 53 | 208 | 21 | 6 | 5 | 3 | 282 |
| Average ratio | 0.41 | 0.44 | 0.29 | 0.48 | - | - | - | 0.35 |
| Min ratio | 0.18 | 0.09 | 0.10 | 0.05 | - | - | - | 0.05 |
| Max ratio | 0.52 | 0.83 | 0.90 | 0.95 | - | - | - | 0.84 |

Table 3. Comparison of concentrations of MMA, DMA, TMA and EA in different sites and seasons, gas and aerosol phase.

| Amine | Gas (ppt$_v$) | Aerosol (ng m$^{-3}$) | Site description | Location | Season | Year | Reference |
|---|---|---|---|---|---|---|---|
| MMA | <DL–8.8 | <DL–61.2 | Rural forest | Finland | Spring-early winter | 2015 | This study |
| | max. ~2 | | Rural forest | AL, USA | Summer | 2013 | You et al. (2014) |
| | 5 | | Semi-rural | DE, USA | Summer | 2012 | Freshour et al. (2014) |
| | 4 | | Rural | OK, USA | Spring | 2013 | Freshour et al. (2014) |
| | 4 | | Urban | MN, USA | Autumn | 2012 | Freshour et al. (2014) |
| | 0.26* | | Urban | Turkey | Summer | 2004-2005 | Akyüz (2007) |
| | 1.3* | | Urban | Turkey | Winter | 2005-2006 | Akyüz (2007) |
| DMA | <DL–4.1 | <DL–55.5 | Rural forest | Finland | Spring-early winter | 2015 | This study |
| | max ~7[a] | | Rural forest | AL, USA | Summer | 2013 | You et al. (2014) |
| | 28[a] | | Semi-rural | DE, USA | Summer | 2012 | Freshour et al. (2014)[a] |
| | 20[a] | | Rural | OK, USA | Spring | 2013 | Freshour et al. (2014)[a] |
| | 42[a] | | Urban | MN, USA | Autumn | 2012 | Freshour et al. (2014)[a] |
| | 2.18* | | Urban | Turkey | Summer | 2004-2005 | Akyüz (2007) |
| | 2.96* | | Urban | Turkey | Winter | 2005-2006 | Akyüz (2007) |
| | <2.7 | <2.7 | Urban | Canada | Summer | 2009 | VandenBoer et al. (2011) |
| | 6.5±2.1 | 0.1±0.2 | Rural | Canada | Autumn | 2010 | VandenBoer et al. (2012) |
| | 42±30[a] | | Rural forest | Finland | May-October | 2011 | Kieloaho et al. (2013) |
| | max 10 | | Urban | GA, USA | Summer | 2009 | Hanson et al. (2011) |
| | | 9.3-20.5 | Semi-arid | AZ, USA | Whole year | 2012-2013 | Youn et al. (2015) |
| TMA | <DL-6.1 | | Rural forest | Finland | Spring-early winter | 2015 | This study |
| | 34–80 | | Rural forest | Finland | Spring | 2002 | Sellegri et al. (2005) |
| | max. ~20[b] | | Rural forest | USA | Summer | 2013 | You et al. (2014) |
| | 6[b] | | Semi-rural | DE, USA | Summer | 2012 | Freshour et al. (2014) |
| | 35[b] | | Rural | OK, USA | Spring | 2013 | Freshour et al. (2014) |
| | 19[b] | | Urban | MN, USA | Autumn | 2012 | Freshour et al. (2014) |
| | 15[b] | | Rural forest | AL, USA | Summer | 2013 | You et al. (2014) |
| | <2.7[c] | <2.7[c] | Urban | Canada | Summer | 2009 | VandenBoer et al. (2011) |
| | ~1[c] | 1±0.6[c] | Rural | Canada | Autumn | 2010 | VandenBoer et al. (2012) |
| | 21±23 | | Rural forest | Finland | May-October | 2011 | Kieloaho et al. (2013) |
| | ≤6.8×10[3] | | Agricultural | CA, USA | Autumn | 2013 | Dawson et al. (2014) |
| | | max 9±7 | Wildfire[#] | Canada | Summer | 2015 | Place et al. (2017) |
| EA | <DL-8.2 | | Rural forest | Finland | Spring-early winter | 2015 | This study |
| | 0.35* | | Urban | Turkey | Winter | 2005-2006 | Akyüz (2007) |

a: Mass 46 i.e. DMA+EA, b: Mass 60 i.e. TMA+PA, c: TMA+DEA, *: Units in ng m$^{-3}$, #: Samples are collected in British Columbia during wildfires

**3.2.1 Monomethylamine**

A spring maximum was observed for MMA(tot) (max. 50 ng m$^{-3}$) and the concentrations correlated with the sum of $NH_3$ and $NH_4^+$ ($R^2$=0.52, Fig. 3). During spring we observed two occasions when MMA(tot) and the sum of $NH_3$ and $NH_4^+$ concentrations increased considerably at the same time. The concentration increase in March is characterized with rain (Fig. 4a) and the later increase in April took place during night with decreasing wind speed and higher temperature (Fig 4b). This increase could be connected to evaporation from melting snow and ground. Bigg et al. (2001) suggest that water from

melting snow penetrate the soil and leaf litter beneath the snow, displacing gases produced by decomposition of organic material. These gases are then released to the air, where they participate in the nucleation process. At humid conditions this bubbling of gases would be efficient, whereas the evaporation to air would be more efficient on warm, sunny days.

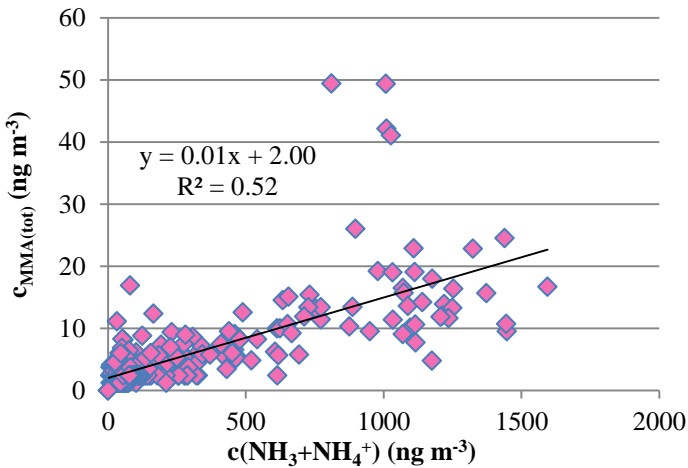


Figure 3. Concentrations (ng/m$^3$) of total MMA vs concentrations of $NH_3+NH_4^+$ in March and April 2015.

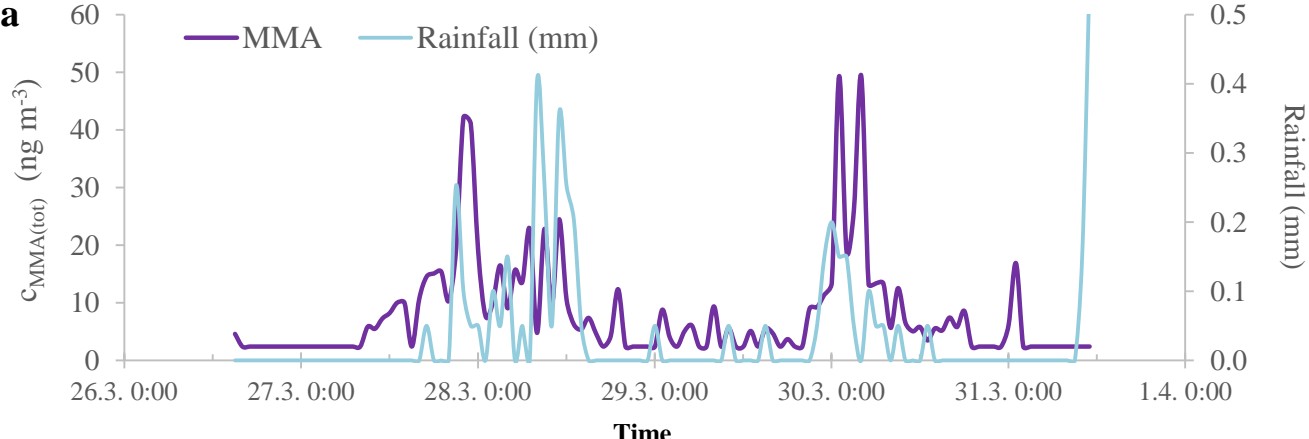


Figure 4: MMA(tot) concentrations and rainfall measured in Hyytiälä during spring 2015 in March (a) and MMA(tot) concentrations, wind speed and ambient temperature in April (b).


Most of the MMA was in aerosol phase (Table 3 and S2): monthly mean of aerosol phase MMA(a) varied between <2.4 and 6.8 ng m$^{-3}$ (Table S3), while monthly means of gas phase MMA(g) were below DL throughout the measurements. In early winter (late November to early December) MMA was not detected. NH$_x$ showed similar seasonal variation as MMA with the maximum in March and lower concentrations towards the end of summer. During spring NH$_x$ was also mainly in aerosol

phase.

In earlier studies (Table 3) You et al. (2014) detected gaseous MMA(g) with CIMS in Alabama forest in summer, at about the same concentrations as our measurements (maximum ~2 ppt$_v$, ca. 3.8 ng m$^{-3}$). Also Freshour et al. (2014) measured

MMA(g) with AmPMS in 3 different sites in USA, and their mean concentrations were at the same level as ours (4–5 $ppt_v$, ca. 5.1–6.4 ng m$^{-3}$). Akyüz (2007) took urban outdoor air samples in Turkey during summer times 2004-2005 and winter times 2005-2006, and analysed them later with GC-MS. MMA(g) mean results were 0.26 and 1.30 ng m$^{-3}$, respectively. Values are at similar levels to our measurements. That is surprising, because in urban area we expect many MMA-sources (e.g. industry and cars, Ge et al. 2011), so higher mean concentrations would have been expected.

### 3.2.2 Trimethylamine

TMA(tot) had higher concentrations in March after which they declined, before increasing again in July to their maximum concentrations suggesting biogenic sources (Fig. 2, Fig. S4). TMA(tot) concentrations also peaked at the end of March during rain simultaneously with MMA(tot) and the sum of $NH_3$ and $NH_4^+$ increasing from about 1.5 to 6.0 ng m$^{-3}$, so melting snow and ground could also be the sources of TMA as discussed in 3.2.1. During summer TMA(tot) concentrations increased again concomitant with the sum of $NH_3$ and $NH_4^+$ in July. The share of the gas phase was roughly half of the aerosol phase concentration throughout the measurements (Table 2 and S2). TMA(tot) did not show a clear diurnal variation (Fig. 5).

Kieloaho et al. (2013) collected filter samples of gaseous amines from the same boreal forest as we did from May to October 2011 and they also measured low concentrations for the sum of TMA(g) and PA(g) in July. In their measurements the concentrations increased during autumn. You et al. (2014) measured gaseous $C_3$-amines (TMA and PA) with CIMS in a forest in Alabama from June to early July in 2013 and their highest concentration (~15 $ppt_v$, ca. 36 ng m$^{-3}$) was ~10 times higher than ours (3.5 ng m$^{-3}$). Dawson et al. (2014) collected TMA-samples in ion resin cartridges from late August to middle September near a cattle farm in Chino, California, and analyzed the samples with IC. Their results varied from 1.3-6.8 $ppb_v$ (ca. 3.1–16.4 µg m$^{-3}$), so they measured ~1000 times higher concentrations than we did. This is not surprising, because cattle are a known source of amines. Sellegri et al. (2014) measured amines in March 2002 with CIMS in same boreal forest that we did. They found TMA(g) with mixing ratios 34–80 $ppt_v$ (ca. 82–193 ng m$^{-3}$), so their results are ~30 times higher than ours. Ambient conditions were different than ours when they measured TMA, and this could be one reason for the higher concentrations they observed.

### 3.2.3 Dimethylamine

DMA(tot) concentrations also increased from about 3 to 6 ng m$^{-3}$ during the MMA episode in April. Moreover, both particulate and gas phase DMA had maximum concentrations in July suggesting a biogenic source (the highest value was 14.5 ng m$^{-3}$ in the aerosol phase and 7.5 ng m$^{-3}$ in the gas phase). The particle fraction was again generally more abundant than the gaseous fraction. Because amines can be expected to partition in the aqueous aerosols (Ge et al. 2010), it is not

surprising to find them mostly in the aerosol phase, considering the high average relative humidity measured (>68%). In August the concentrations decreased, and they were the lowest during early the winter. Kieloaho et al. (2013) measured also high gas phase concentrations of the sum of DMA and EA in July, reaching a maximum of ~75 $ppt_v$ (ca. 138 ng $m^{-3}$). In their measurements the concentration levels decreased in August similar to our measurements. High DMA and TMA

concentrations in summer could indicate biogenic sources. However, these amines concentrations did not correlate with monoterpene concentrations like EA (see section 3.2.4). VandenBoer et al. (2011) measured both DMA(g) and DMA(a) with AIM-IC from late June to early July 2009 in an urban area, with highest concentration of 2.7 $ppt_v$ (ca. 4.6 ng $m^{-3}$) and 2.7 ng $m^{-3}$ which were at the same level as our DMA(g) in July (7.5 ng $m^{-3}$). Hanson et al. (2011) also measured DMA concentrations with AmPMS in an urban area with a little higher gas phase concentrations (maximum of 10 $ppt_v$, ca. 19 ng

$m^{-3}$) than in the studies mentioned earlier. Ge et al. (2010) gives DMA also urban sources (e.g. tobacco smoke, automobiles), so that can explain results from Hanson et al. (2011). Youn et al. (2015) measured DMA aerosols and cloud water, and they noticed that DMA concentrations in PM1-aerosols peaked in September. We were also expecting high concentrations in autumn, but due to instrumental problems we unfortunately missed the season. In July we measured from PM10 particles the average concentration of 8.4 ng $m^{-3}$, and Youn et al. (2015) measured from PM1 particles about twice as high concentration.

Different measurement sites could explain the difference. Youn et al. also noticed that DMA(a) displays a unimodal size distribution with dominant peak between 0.18 and 0.56 μm, and concluded that it indicates aminium salt formation with sulphate..

In August, DMA(tot) had a diurnal variation with a daytime maximum (Fig. 5), but during some nights the concentrations

also increased slightly. The DMA(tot) afternoon maxima could be caused by re-emission of DMA that has earlier deposited on surfaces and evaporates when temperature increases during the afternoon. The maximum could also be related to direct biogenic emission. Usually ambient concentrations of biogenic volatile organic compounds, which have temperature dependent emissions, peak during night-time due to weak atmospheric mixing and lack of hydroxyl radical reactions which only take place during daytime (Hakola et al. 2012). The concentrations of light dependent BVOC emissions such as

isoprene have daytime maxima because they are emitted only during daytime. Thus, the DMA source could be light dependent. DMA(tot) peaks also at night. Because the atmospheric mixing in the night is weak and there are no OH-reactions, even small emissions can be trapped in a shallower atmospheric boundary layer and cause the increase in concentrations.


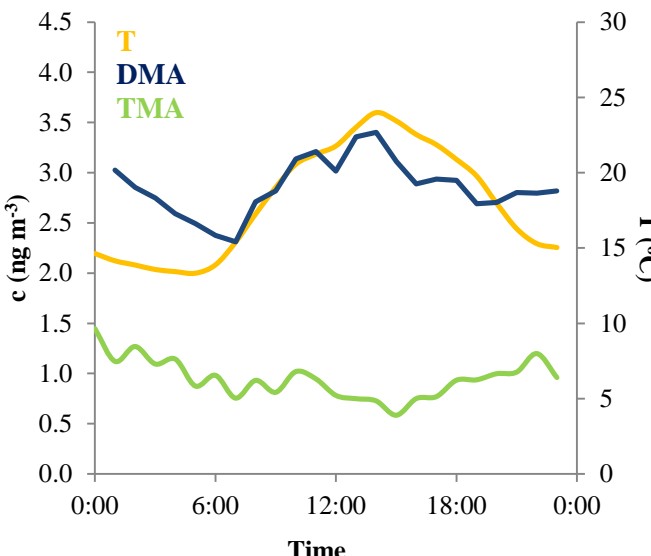

Figure 5. Mean diurnal variation of total DMA (blue), total TMA (green) concentrations and temperature (yellow) in August 2015.

### 3.2.4 Ethylamine

EA(tot) concentrations were low throughout the measurements, but showed a clear diurnal variation in July with a maximum at night (Fig. 6). Monoterpene concentrations were measured simultaneously at the same site and had similar diurnal pattern. This type of diurnal variation is typical for many reactive compounds having local sources in boreal forest (Hakola et al. 2012). Low daytime concentrations are due to strong atmospheric mixing and reactions with OH radicals. The rate coefficients of alkyl amines are slightly lower, but comparable to monoterpene reactions with OH radical. The most common monoterpenes in the boreal forest are α-pinene, 3-carene and β-pinene (Hakola et al. 2012). Their rate coefficients for reaction with OH are $53.7 \cdot 10^{-12}$, $88 \cdot 10^{-12}$, and $78.9 \cdot 10^{-12}$ cm$^3$ molecule$^{-1}$ s$^{-1}$, respectively (Atkinson 1994), whereas MMA, EA, DMA and TMA rate coefficients with OH are $22.26 \cdot 10^{-12}$, $29.85 \cdot 10^{-12}$, $65.53 \cdot 10^{-12}$, and $69.75 \cdot 10^{-12}$ cm$^3$ molecule$^{-1}$ s$^{-1}$, respectively (U.S. EPA, 2017). Similar diurnal patterns and reactivities indicate that EA has a biogenic source. Kürten et al. (2016) measured $C_2$-amines (i.e. DMA and EA) with CI-APi-TOF in Germany near 3 dairy farms and forest from May to June 2014. They did not observe clear diurnal variation for $C_2$-amines. In our measurements, EA and DMA had opposite diurnal variations (see section 3.2.3). That could be the an explanation for the observations of Kürten et al. (2016), where both $C_2$-amines were measured together. Aküez (2007) measured EA(g) 0.35 ng m$^{-3}$ (mean concentration) in an urban area in Turkey during winters 2005-2006, and the concentrations were at the same level as ours.


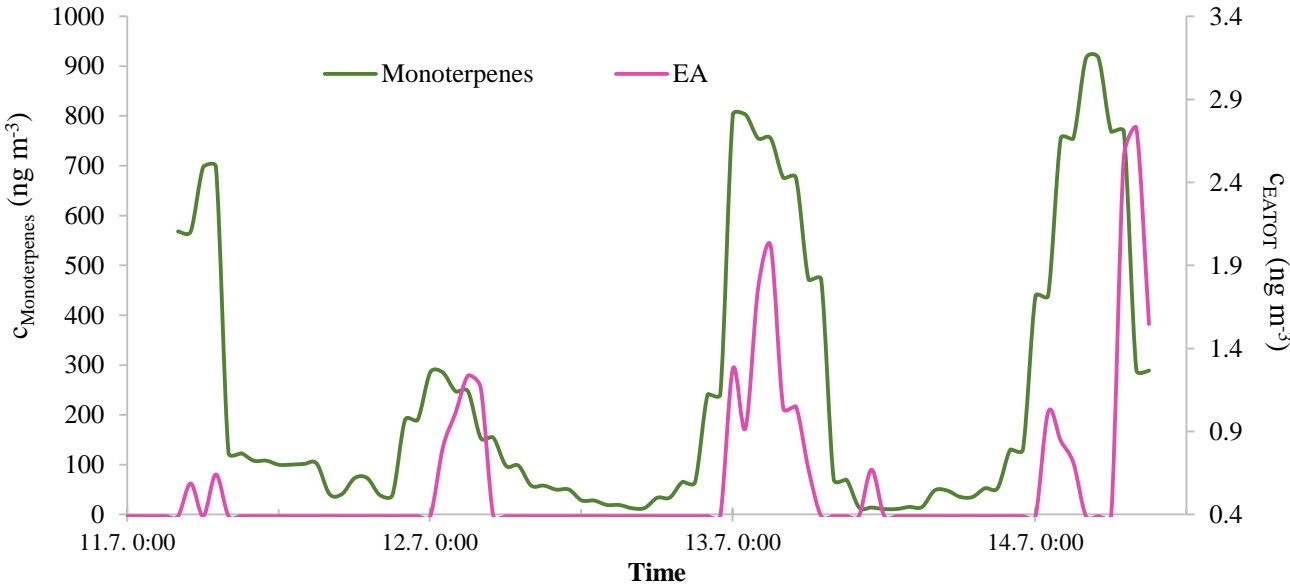

Figure 6. EA(tot) and monoterpene concentrations in Hyytiälä in July 2015. The EA(tot) concentration axis starts from 0.36 ng m$^{-3}$, because values under that are below the detection limit.


### 3.3 Correlations between meteorological quantities and amines

We noticed that the concentrations of DMA(g) followed, although vaguely, the variations of both air and soil temperature (Fig. S5), so it was reasonable to study whether there are any clear relationships between the amine concentrations and parameters describing ambient conditions. Wecalculated linear regressions of amines, ammonia and ammonium vs. air relative humidity (RH) and temperature (T) as well as soil temperature (ST) and soil humidity (SH). The results of the linear regression analyses of the amines, ammonia, ammonium, and the ambient conditions are presented in Tables S4 and S5 for the gas and aerosol phase, respectively.


In the gas phase DMA had the strongest correlation with ambient condition parameters, suggesting that DMA(g) concentrations increase with increasing air temperature, soil temperature and soil humidity but decrease with increasing atmospheric humidity and wind speed. The scatter plots of DMA(g) vs these parameters (Fig. 7) shows, however, that the relationships are different in different seasons. The most consistent relationships of DMA(g) are with air and soil temperature, the slopes of the linear regressions are positive for the whole data and for summer alone (Fig. 8).

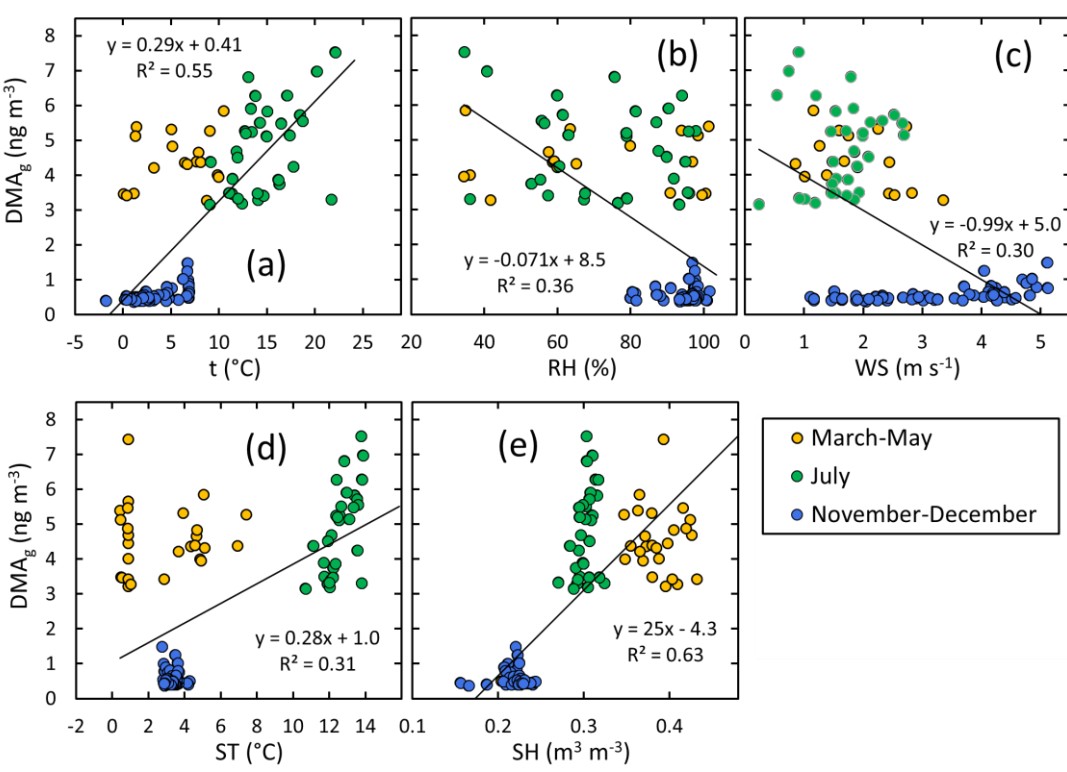


Figure 7. DMA in the gas phase vs selected ambient condition parameters: a) air temperature, b) relative humidity, c) wind speed, d) soil temperature, and e) soil humidity in spring, summer, and early winter. The linear regressions shown in the plots were calculated using the data points of all seasons.

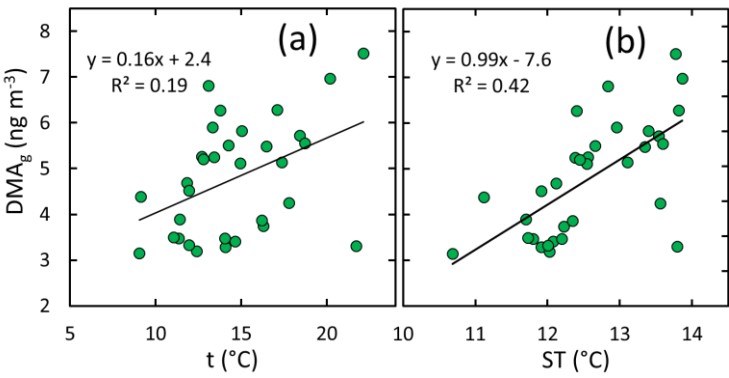


Figure 8. DMA in the gas phase vs a) air temperature, b) soil temperature in summer.

In the gas phase the second strongest correlations – even though weak – are those of TMA against environmental conditions (Table S4). Interestingly, when looking at all data, TMA(g) concentration seems to decrease with increasing air and soil temperature (Fig. S6), opposite to the relationship of DMA and temperature. As already mentioned TMA concentrations were high in spring and they are likely to originate partly from melting snow and ground, whereas DMA might have biogenic sources in summer, which could explain different correlation behavior. The scatter plot of TMA(g) vs. temperature
(Fig. S6) also reveals that the relationship is not consistent in all seasons: in summer it is even a vaguely positive, statistically not significant positivel relationship. The ammonia concentration increased with air temperature consistent with Makkonen et al. (2014) and decreased with increasing relative humidity. The latter suggests that at high humidity surfaces are moist and ammonia gets absorbed onto the water.

All the gas phase amines except MMA were found to have a positive correlation with soil water content. The studied amines are water soluble and therefore negative correlation would be expected if the soil would act only as a sink. However, our results suggest that soil processes are producing amines and they may be enhanced with increasing humidity. Forest soils are a reservoir of the alkyl amines (Kieloaho et al. 2016) and modelling studies have shown that they can act as a source of alkyl amines to the atmosphere (Kieloaho et al. 2017). With their model Kieloaho et al. (2017) found a positive correlation with
soil temperature for soil-to atmosphere flux of DMA, but correlation with soil water content was opposite to our observation.


### 3.4 Correlations of amines with nano-particle concentrations

In addition to the dependency of amine concentrations on ambient conditions, the relationships between aerosol number and amine concentrations were studied with a similar regression analysis. The amine concentrations were compared with the total number concentration integrated from the size distributions measured with the DMPS ($N_{tot}$), with the aerosol number concentrations in the size ranges 1.1-2 nm and 2-3 nm, measured with the PSM ($N_{1.1-2nm}$ and $N_{2-3nm}$, respectively) and with the aerosol particle and cluster number concentrations between 3 and 25 nm measured with the DMPS ($N_{3-25\ nm}$). The regression analysis results for the gas-phase amines and aerosol phase amines are presented in Tables S6 and S7, respectively.

The period during which both the MARGA-MS detected DMA(g) concentrations above the detection limit and the PSM detected cluster-mode aerosols simultaneously was short. There were 33 data points for the regression analysis. There was a weak positive correlation between them (Fig. 9) even though the correlation was statistically not significant ($R^2 = 0.06$, p = 0.18, Table S6). The correlation had some dependence on the ambient conditions: air relative humidity (RH) and temperature (T) as well as soil temperature (ST) and soil humidity (SH). The correlation was more significant when both soil and air were humid (RH > 90 %, SH > 0.3 $m^3$ $m^{-3}$). The linear regression calculated by using only those data that were measured RH>90% has a higher correlation coefficient and slope is statistically significant ($R^2 = 0.63$, p = 0.006, Table S6, Fig. 9b) but it has to be noted that there were only 10 simultaneous data points at the high RH.

There was no correlation between the slightly larger aerosols ($N_{2-3nm}$) and DMA(g) (Table S6), suggesting that DMA(g) took part in the initial steps of secondary aerosol formation namely clustering. This is qualitatively in agreement with an experimental CLOUD chamber study where it has been demonstrated that even very small amounts of DMA greatly enhance the formation of nano-particles (Almeida et al. 2013, Lehtipalo et al. 2016). In the aerosol phase DMA was the only amine that had a statistically significant correlation with the cluster-mode number concentrations and as for the gas-phase the correlation coefficient was higher at high relative humidity (Table S7). Other ambient condition quantities apparently did not affect this relationship (Fig. 9).

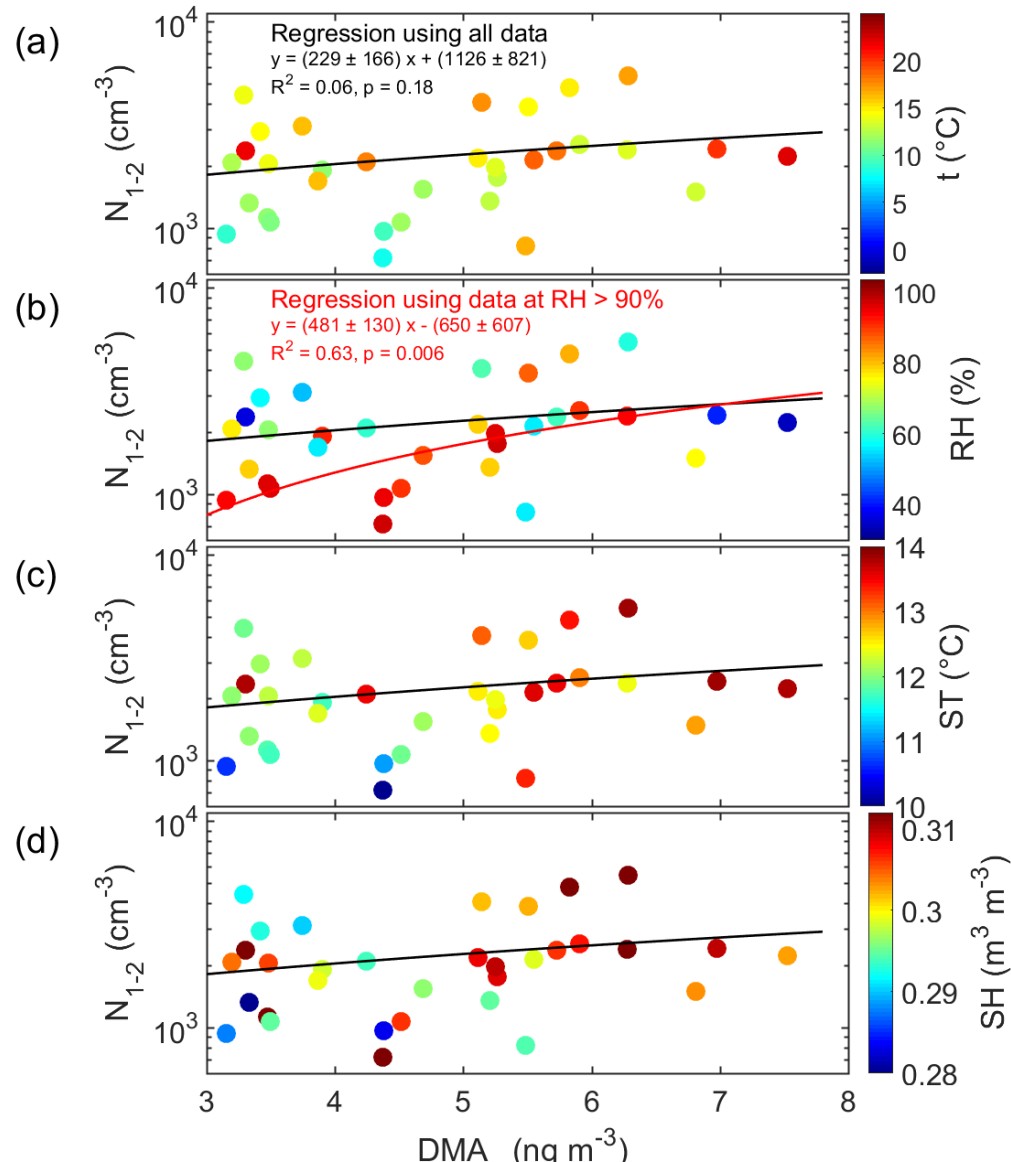

Figure 9. Cluster-mode aerosol number concentration ($N_{1.1-2nm}$) as a function of dimethyl amine (DMA) concentration in the gas phase and color-coded with a) air temperature (T), b) air relative humidity (RH), c) soil temperature (ST) and d) soil humidity (SH). In all subplots the black line shows the linear regression calculated by using all data and in b) the red line shows in addition the linear regression by using only those data that were measured at RH > 90%.



There were considerably more simultaneous data points of the cluster-mode aerosol number concentration and ammonia (NH$_3$). The correlation N$_{1.1-2nm}$ vs. NH$_3$ was statistically significant (R$^2$ = 0.13, p<0.001, Table S6). In addition, this
correlation apparently also depended on the ambient conditions so that in warm (T >15°C, ST > 14°C) and dry (RH < 60%, SH < 0.25 m$^3$ m$^{-3}$) conditions the positive correlation was more obvious (Fig. 10). In the aerosol phase ammonium (NH$_4^+$) did not correlate at all with the cluster mode particle number concentration but positively with the total number concentration (Table S6) as expected. The other amines did not have any significant correlations with the aerosols in the smallest aerosol size ranges.


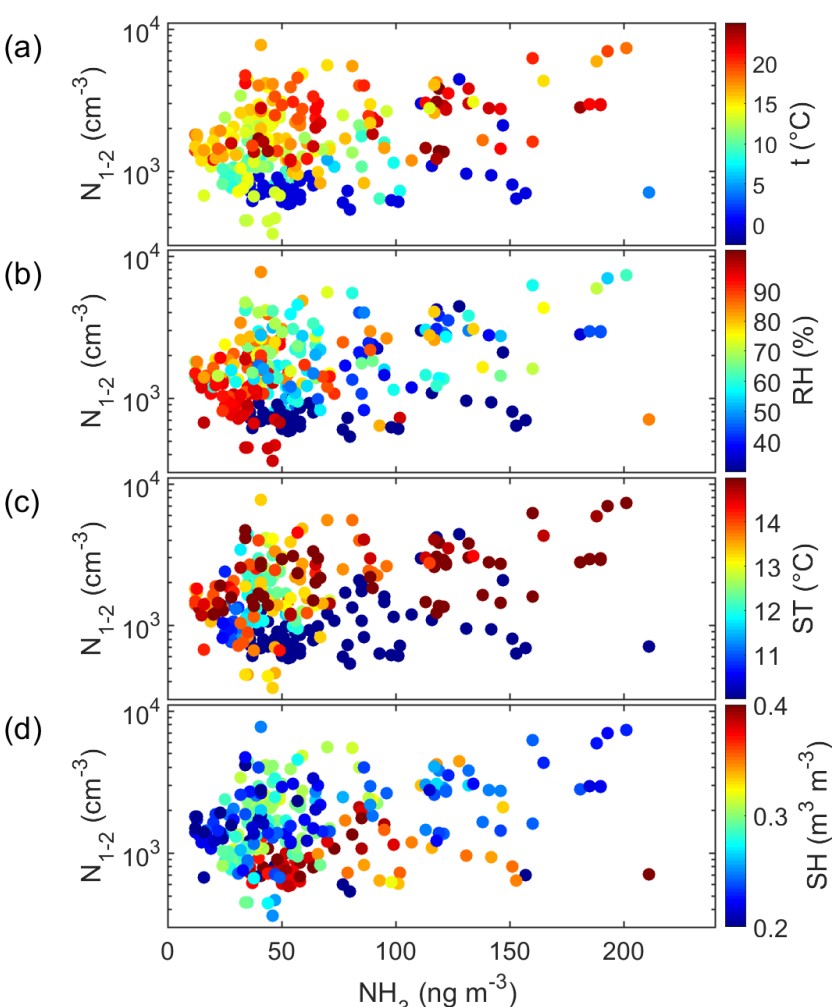

Figure 10. Cluster-mode aerosol number concentration (N$_{1.1-2nm}$) as a function of ammonia (NH$_3$) concentration and color-
coded with a) air temperature (t), b) air relative humidity (RH), c) soil temperature (ST) and d) soil humidity (SH).

## 4. Conclusions

An on-line method using in-situ ion-chromatograph with mass-spectrometric detection for measuring amines in low concentrations from the ambient air both in the gas and aerosol phase was developed. In situ amine and ammonia measurements were conducted in SMEAR II station (Hyytiälä, Finland) from March 2015 to December 2015, covering altogether about 8 weeks. Concentrations of 7 different amines and ammonia in aerosol- and gas-phase were measured with 1-hour time resolution.

The developed MARGA-MS method was suitable for field measurements of amines. The DLs were low (0.2–11.4 ng m$^{-3}$), and the accuracy and precision of IC-MS analysis were moderately good. With the method amines with same masses or same retention time were separated, only DEA and BA were incompletely separated. However, MARGA-MS had some technical drawbacks (e.g. consumption of ~40 l of solutions per week).

The amines turned out to be a heterogeneous group of compounds; different amines are likely to have different sources. All amines had higher concentrations in the aerosol phase than in the gas phase. MMA and TMA concentrations were the highest in spring concomitant with ammonia and ammonium. Melting of snow and ground can be the source of these compounds. The decomposing litter and organic soil layer beneath snow can release organic compounds to snow cover and to the atmosphere.

TMA had an additional maximum simultaneously with DMA during summer, and EA was only detected in July. The summer maxima could indicate biogenic sources. However, unlike EA, DMA and TMA did not show similar diurnal variation as monoterpenes. The diurnal variation is determined by the balance between emissions, reactivity and mixing in the atmosphere. Usually ambient concentrations of biogenic volatile organic compounds, which have temperature dependent emissions, peak during nighttime due to inefficient mixing and lack of hydroxyl radical reactions which only take place during daytime. The missing daytime minima of DMA and TMA can be due to light dependent biogenic source, or TMA and DMA might be re-emitted from surfaces during daytime, when temperature increases.

All amines except MMA correlated positively with soil humidity, which could indicate a humidity dependent production mechanism. Gas-phase DMA correlated positively with small 1.1-2 nm aerosols, when both soil and air were humid. It did not correlate with slightly larger aerosols at all, suggesting that gas phase DMA may be important in new aerosol formation.

*Data availability.* The datasets can be accessed by contacting the corresponding author.

*Competing interests.* The authors declare that they have no conflict of interest.

*Acknowledgements*. The financial support by the Academy of Finland Centre of Excellence program (project no 272041) and
       Academy Research Fellow program (project no 275608) are gratefully acknowledged.

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
