# Peer review of "Amines in Boreal Forest Air at SMEAR II Station in Finland"

_Atmospheric Chemistry and Physics, 2017_

## Referee Comment (RC1) · Anonymous Referee #1 · 14 Nov 2017

The manuscript "Amines in Boreal Forest Air at SMEAR II Station in Finland" provides an in-situ observation of 7 amine species in both particle and gas phases along with ammonia and ammonium ion over a continental rural area at the Finnish boreal forest site, HyytiaÌĹlaÌĹ, in year 2015 from March to December. The study lasted for a total of 8 weeks, spread out over 8 months. It applies a newly developed measurement technique for amines that combines online ion chromatography and an electrospray ionization quadrupole mass spectrometer. The data analysis relies on simple linear regression to explore the relationships between amines and several environmental factors including rainfall, soil temperature, soil moisture, ambient air temperature and ambient air relative humidity. In addition, the work is used to explore amine species, especially dimethylamine in new particle formation through nucleation process. The authors pro-

vide considerate insights of diurnal and seasonal variations of amines over the study region, and highlight the different production mechanisms and sources among the detected amines.

The topic of this paper is relevant to the journal and has important scientific contributions to the knowledge of amines in remote continental area, especially with relative longer period observations compared with previous studies. The experimental design is good. However, improvements are needed in the Results section, especially for the quality of figures and tables, in order to deliver to readers more concise and better visualized results. Also, authors should provide more thoughtful interpretations before drawing conclusions.

Prior to publication, the authors should address the specific comments below.

1. Please provide detailed information of the sampling period. What was the rationale to pick the 8 weeks during the 8 months? Since the study emphasizes seasonal variations, how confident can one be with measurements from relative short sampling periods in each month to make conclusions about seasonal changes?

2. Section 2.2: Authors simply use one sentence to cite previous work as Junninen et al. 2009 without a brief description of what this portal is. A bit more information is warranted. In addition, Junninen et al. 2009 is missing in the reference list. Please check and add in.

In table 1, do environmental conditions have small or big variations during each month? please add standard deviations to each mean value. Also, it would be helpful to make statements of diurnal changes (i.e. day vs. night). Also, please provide information about rain, soil moisture, and soil temperature, as they are important environmental factors in the discussion.

3. Misleading description at the very beginning of section 3.1: "Figure 1 shows the monthly means and medians of total amine concentrations (sum of gas and aerosol

phases) ". Figure 1 only shows means. Correction is needed.

It is confusing to claim monthly mean changes as seasonal variations (shown in figure 1) unless the authors define the seasons at first. In the figure, half of the species (EA, DEA, PA and BA) have different scales than the rest. Please consider using two different y-axis scale in one plot or having two separate plots in order to provide more clear trends for each species.

Please clarify the meaning(s)/significance of showing the sum concentrations of gas and particle phases measurements (Figure 1 and 2). Tables 3 and 4 seem to deliver similar cumulative results as Figures 1 and 2 but in separate phases, which are arguably better to understand.

4. Line 185-190: "The concentration increase in March is characterized with rain (Fig. 4) and the later increase in April took place during night with decreasing wind speed and higher temperature. This increase could be due to evaporation from melting snow and ground."

In Figure 4, the time scale on x-axis is too rough to provide clear vision of diurnal variation. Improvement is needed.

It indicates rainfall is featured with high MMA concentration in March (Figure 4), which is mostly in the particle phase, as shown in Table 4. Does such high MMA relate to previous cloud processing? Except rainfall, do the other environmental conditions have potential influences? Authors should expand discussion here.

No detailed information of wind speed and ambient temperature is provided to support the discussion. More explanations and possible references could assist the discussion about evaporation from melting snow and ground.

5. For Section 3.1.4, the authors should include discussion to more extensive literature examining this species. There are a number of references in fact for all of the amines, but for DMA, at the minimum a couple of references with discussion of sources and

behavior of DMA are the following:

Youn, J. –S., et al. (2015). Dimethylamine as a major alkyl amine species in particles and cloud water: observations in semi-arid and coastal regions, Atmos. Environ., 122, 250-258, doi:10.1016/j.atmosenv.2015.09.061.

Murphy, S. M., et al. (2007). Secondary aerosol formation from atmospheric reactions of aliphatic amines, Atmos. Chem. Phys., 7, 2313–2337.

6. Figure 5: Time scale on x-axis is too rough to tell diurnal circle. Improvement is needed.

7. Section 3.1.4: Authors emphasize DMA (and TMA) concentration higher in summer (i.e. August) due to biogenic sources. However, interpretations/discussion leading to that conclusion are not convincing in my opinion. It mentioned that DMA does not show correlation with biogenic tracer such as monoterpenes, while isoprene is noted as having light dependent emissions. Please provide supportive BVOC tracer information if the data is applicable. Are there BVOC tracers other than isoprene found related to variations of DMA and (or) TMA?

In Figure 6, DMA shows strong diurnal cycle while TMA doesn't. Is the DMA diurnal circle found only during summer, especially August? Why is it that TMA does not have such a strong diurnal circle as it also mentioned in section 3.1.2? Authors should expand discussions here.

8. Figure 7: The current plot is hard to show the clear relationship between DMA and selected environmental factors, especially for data around July and December.

Please consider zooming in time scale on x-axis, for an example, using a discontinued time series.

9. Line 295- 300: Does TMA negatively correlate with ambient temperature consistently or is this sensitive to season?

Minor comments

1. Authors should consider adding a site map in section 2.1 in order to provide readers visualized information of study area.

2. Line 115: Why are the DL calculation methods for DMA and TMA different from the rest? In table 2, DMA shows different DLs in two time periods, while TMA doesn't. Please clarify the reason(s).

3. Table 1: What is the difference between "mean" and "average"? If they are same, please be consistent.

Grey shade is not necessary if the color does not have meaning. Same comment applied to Table2.

4. Table 2: Some species have a comma after their names while the others do not. Why NH3 and NH4+ are not mentioned in gas (particle) phase as the rest? Keep consistent style please.

5. Table 4: Typo for DMA median value (particle phase) in July. "4,9" should be "4.9".

6. Figure 4: Add label for x axis. Same comment for Figure 5. The x-axis represents dates, but it is unclear. In contrast, Figure 7 has better x axis format. Please be consistent in plot style.

7. Figure 7: Units on y axis should be in parentheses.

---

## Referee Comment (RC2) · Anonymous Referee #2 · 16 Nov 2017

**General Comments**

The Authors present several weeks of ambient observations of seven atmospheric alkyl amines in the gas and particle phases using the MARGA system interfaced with a quadrupole mass spectrometer. The quantities of these amines from each of the observation periods is reported and trends discussed from a variety of perspectives (e.g. diurnal, inter-monthly, etc.) They investigate a variety of correlations of the measured amines with accompanying physical observations made at the same location. Overall, this manuscript is not suited for publication in Atmospheric Chemistry and Physics without many major issues being addressed.

**Major Comments**

[Figure]

1. There are far too many data tables and figures that are irrelevant to the structure of the results and discussion. Most can be replaced with a single sentence. These detract from the quality of the observations and should be relocated to a Supporting Information document. These are noted in detail below.

2. The Authors claim that they are reporting the longest time series of amines measurements to date, but the measurements are short-duration periods made in different months. A time series implies continuous data collection and the authors should revise the manuscript to be clear that they are reporting eight weeks of observations from different months over the course of a year from the same observation site. Further to this point, the sampling strategy reported and the findings discussed in the paper are all weakened because of the intermittent nature of these observations. The limitations of the dataset need to be presented clearly. Figure 7 is the only depiction of the full measurement time series and it appears that even within each observation period that there are gaps in the data which are not clearly explained. How can the Authors justify their conclusions regarding monthly/seasonal trends if they do not measure continuously throughout each?

3. The manuscript does not appear to have a clear purpose or objective. There are several discussions made throughout the manuscript that are not joined in a clear narrative, the final paragraph of the introduction for example, which confuses the meaning and scientific contribution of the findings.

a. The Authors present an advancement in atmospheric amine measurement capabilities through the use of an ion chromatography-mass spectrometry system, but do not clearly demonstrate the necessary performance metrics (e.g. a sample chromatogram demonstrating the ability to speciate the suite of analytes likely to be encountered in the atmosphere). The data then presented in the tables and figures is still largely below the instrument detection limits (e.g. Table 4), so is this improvement really meeting the observational needs of the research community?
b. A timeseries of diurnal observations and linkages to known boreal biogenic processes is discussed for some amines.

c. The relationship of gaseous DMA to a variety physical observations is made, concluding that new particle formation is higher when both DMA and RH are high.

To me, this manuscript is a first long-term survey of sources and phase-distribution of amines at the sub-pptv level in a remote boreal forest environment using supporting physical measurements to initiate a better understanding of this entire class of compounds relative to what we already know about ammonia. The Authors should carefully review the findings of their work and convey the purpose of their work clearly as it will also strengthen the structure of their results and discussion. The introduction should be subject to a major revision based on the determined purpose and presented structure of the resulting manuscript as it currently does not do so.

4. The methods section is not detailed enough to evaluate whether the DLs for the MARGA-MS are robust and reliable. There are many issues here that need to be addressed that are detailed in the Technical Comments below.

5. The manuscript has many typographical and technical errors that should have been addressed prior to submission (e.g. use of the term 'aerosol particles'). The Authors are strongly encouraged to seek external review of their work by peers and colleagues after revision prior to resubmission for further review.

Technical Comments

The following comments are not an exhaustive list of the corrections required for this manuscript to be acceptable for publication. The Authors, in addressing the major comments, will likely correct much of the unmentioned issues under their own revision.

Page 1, Line 15: The Authors state that they can separate and detect 7 different amines, but do not show any evidence of this performance, nor does the literature cited for the methodology used. If there is prior work demonstrating the quality of this

method performance for amines, specifically, it should be cited and briefly summarized. If not, then the Authors are missing an opportunity to present a significant advance in the online measurement of atmospheric amines simultaneously in the gas and particle phases.

Page 1, Line 17: The term 'possibly' is used here, which is speculative. Such conclusions, while potentially acceptable for the discussion, should not be present in an Abstract. Major findings with solid support only should be presented here.

Page 1, Lines 21-22: There is no conclusive evidence presented for DMA and TMA biogenic sources. This is only true for EA.

Page 1, Lines 23-24: What is the purpose of presenting the means and medians for these measurements in the Abstract? This does not seem necessary.

Page 1, Line 27: '0.63 EA' This is an example of missed typographical errors that should be identified prior to manuscript submission.

Page 2, Line 34: The amine class of compounds can be more broadly defined. R- can be used to represent both H- and alkyl- substituents.

Page 2, Lines 35 and 38: 'aerosol particle' is incorrect terminology. These are interchangeable terms and typically one or the other is chosen for use throughout a manuscript.

Page 2, Lines 44-45: 'and' is used twice in a row. Another example of missed typographical error.

Page 3, Lines 67-68: The Authors have not presented any information on the range of amines that have been measured, or are even estimated, to be present in the boreal forest prior to this sentence evaluating the potential utility of prior published methods. Depending on the chosen purpose of the manuscript, the Authors should either expand on the findings of previous boreal measurements, focus on the performance of the measurement technique relative to other reports, or link these two themes with a

motivation of greater breadth.

Page 3, Line 89: Why are the exact dates for the measurements not given? The datasets, such as Figure 7, clearly show that these periods were not subject to continuous observation and the details should be provided here.

Page 4, Table 1: Move this to the SI and add soil humidity and soil temperature details to this section of the methods since they don't appear anywhere, but appear regularly throughout the results and discussion.

Page 4, Line 103: Define the MARGA acronym first and then put the acronym in brackets.

Page 4, Lines 105-106: This is far too brief of a description for the interfacing and operation of a mass spectrometer to an ion chromatograph. Does the MARGA suppress the IC eluent prior to analyte measurement? Were conductivity measurements made prior to, but in series, with the mass spectrometer? What solvent and what ratio to the IC mobile phase was added prior to the electrospray? What were the desolvation and transmission settings of the mass spectrometer?

Page 4, Lines 112-113: Was the separation an isocratic or gradient separation? What was the time required for separation? What were the dimensions of the analytical column and what was the particle size of the stationary phase?

Page 4, Lines 114-116: Are these the DLs for the IC system or for the MARGA? That is, were the S/N = 3 calculated from blank injections, the measurements made in 'blank-mode', or from MARGA solvents analyzed when the inlet was overflowed with zero air? Given the need for very sensitive measurement capabilities outlined in the introduction, the Authors should present a more detailed description of how the DLs were determined and the quantitative metrics evaluated to conclude that they are reliable and robust representations of the capabilities of the MARGA. Characterizations of these parameters for IC systems measuring amines have been previously described

by Erupe et al. (2010), Dawson et al. (2014) and Place et al. (2017), those with atmospheric interfaces by VandenBoer et al. (2011, 2012), and IC coupled to mass spectrometry by Verriele et al. (2012)

Page 5, Table 2: This belongs in the results section of the manuscript along with the remainder of the analytical performance metrics. The Authors could consider reporting the DLs as ng/mˆ3 for the particle channel and pptv for the gas channel to improve the clarity of this table.

Page 5, Lines 122-125: How was the d10-DEA introduced to the MARGA-MS as an internal standard? Was it added to the solvent in the particle and gas channels, to the post-suppressor organic solvent, or only to standards that were injected offline? What is the purpose of using this internal standard and how did it perform for the set of reported field measurements? Presumably it was used to track the spray ionization efficiency on an ongoing basis in the post-suppressor organic solvent, but this is not clear. The Authors should also indicate how the internal standard was utilized in back calculating the quantities of all the measured amines and whether it reduced uncertainty in the measurements and by how much.

Also, why is are the calibration standard values listed as ' $\sim$ ' which means approximate? Further to this, the calibration range seems to be far above the observed values, which are typically below 10 ng/mˆ3, which is the lowest calibration standard, and is therefore extrapolating the calibration below the determined range. The Authors should provide proof that the system sensitivity response from 0.1-10 ng/mˆ3 is the same as from 10-300 with at least a 3-point calibration for all of the analytes. They should use this information to determine their measurement accuracy and precision as well, which is not evaluated.

Page 5, Lines 128-129: Since the MARGA was not operated continuously for the analysis of amines, it is important to specify whether the instrument blank values were collected before, during, or after the periods when continuous monitoring was being

performed. Again, I presume this was performed immediately before, intermittently during, and immediately after each observation period so that backgrounds could be corrected throughout each observation period, but the explicit information needs to be presented in the manuscript.

Page 5, Lines 131-133: If the DMA backgrounds are different due to cycling between sampling syringes, it should be possible to explicitly assign a background correction to samples collected with each set of syringes instead of averaging, which will decrease both accuracy and precision of the measurement.

Page 6, Line 135: This should be the start of a new methods section that describes the DMPS measurements.

Page 6, Line 149: Regression calculations can allow some insight into the physico-chemical nature of the amines, but the results and discussion do little to explore the reasons why the variable the authors chose were investigated. What are the chemical and physical mechanisms that may be acting to release amines in the boreal environment? Is there a precedent from laboratory work or prior observations? Given those parameters explored, the working hypothesis seems to be testing whether there is similar release and exchange of amines in the boreal as might be expected given the extensive literature on ammonia. If this is the case, this reasoning should be emphasized throughout the manuscript and supported by citing the relevant literature.

Page 7, Lines 167-168: 'amines were mainly in the aerosol phase' - the averages appear very close to 0.5 and likely are if the variability in the data is considered. The range of the gas fraction values presented in Table 3 suggests this is the case. It would be more accurate to say that the gas fraction was variable, with an average and standard deviation given for the entire dataset.

Page 7, Figure 1: Move this to the Supporting Information. It repeats all of the data presented in Figure 2 and since the observations do not span the entire month, it may be more accurate to add the observation dates to the bar labels in the legend. It would

be more informative for the authors to present some continuous data in a figure that includes many of amines, ammonia/ammonium, the uncertainty in each measurement, and the DLs so that the quality of the MARGA-MS data can be ascertained.

Page 8, Figure 2: How many points are off scale in each of these panels? Why are the DEA measurements suggesting that there were negative quantities detected? Where are the November and December measurements for NH3+NH4?

Page 8, Table 3: Please check that the table formatting is done according to the guidelines of ACP

Page 9, Table 4: Move this table to the Supporting Information and replace with 2-3 simple sentences in the text. Why are both the mean and median values in this table? What does a difference between these two metrics tell us about the amine measurements and why? This table suggests that even MARGA-MS does not have adequate DLs for the boreal environment and the authors should comment on this in the discussion.

Page 10, Figure 3: Remove '(y-axis)' and '(x-axis)' from the caption. This is obvious. Consider a more descriptive caption.

Page 10, Figure 4: This date format is not consistent with previous figures, and the notation is not defined via the axis label. The data presented in this figure are clearly non-continuous between months and goes back to the points above regarding statements by the Authors suggesting that the dataset is continuous when it is not.

What is the purpose of plotting rainfall on this figure? The discussion speculates on 'evaporation from melting snow and ground', but rainfall does not describe either of these processes. What is the physical or chemical rationale for this speculation? Is there precedent in the literature to support this?

Pages 11 -18: The Authors compare their measurements for each species to those from other reports throughout. This would be more easily conveyed through the use

of tables that present the data from this work in comparison to the findings of others, listing relevant parameters such as rural and urban settings.

Page 11, Line 210: Why is TMA discussed before EA, but presented in Figure 6 following the description of EA in Figure 5? Please reorganize the discussion so that figures appear in the order that they are discussed.

Page 12, Figure 5: Most of the observations for EA are below the DL, so why is the line in the figure continuous? It would be more appropriate to add another trace here that denotes the cut off for the DL and to leave gaps in the EA dataset where the measurement is below the DL.

Page 13, Lines 265-266: Soils and surfaces in the boreal are acidic when measured in bulk, so it seems unlikely that deposition and re-emission is a plausible line of speculation. For example, a comparison using compensation point theory for ammonia in these environments could suggest that deposition should be the final fate of bases at the surface, and could support a similar case for the amines since they are stronger bases than ammonia. This would be strong evidence against deposition and desorption cycles and indicate other mechanisms of emission (e.g. decomposition of organic matter).

Page 13, Figure 6: Why did the Authors not explore the diurnal nature of TMA in July when the measured mixing ratios were highest? This compound is known to be released during the flowering of many plants to attract pollinators and should be discussed for context.

Page 14, Line 279: 'concentrations of DMA vary with temperature' - Figure 7 does not demonstrate such a dependence, but the statistical findings in Table 5 do. There is no clear dependence to the eye in Figure 7 between DMA and air temperature.

Page 14, Figure 7: These panels are not alphabetically labeled for reference in the caption. Previous figures suggest there were DMA measurements made in August,

but they are not shown on this figure. Why is this? The findings in Table 5 suggest that the correlations between DMA and a number of parameters are worth noting and those plots would be more valuable than this figure. Consider replacing Figure 7 with a multi-panel figure showing these relationships and the regression statistics from Table 5.

Page 15, Table 5: There is very little here that is meaningful to the discussion. It can be replaced with the figure noted in the previous comment and a couple sentences in the text. Move to the SI or consider removing from the manuscript entirely.

Page 17, Table 6: Same comment as Table 5.

Page 19-22, Figures 8 and 9, Tables 7 and 8: Only depict and present the most meaningful results. That is, present the figure panels that convey information central to the discussion which cannot be easily replaced with three or fewer sentences in the text. In this case, consider combining the most important findings from Figures 8 and 9 into one figure. Move the tables to the Supporting Information or consider removing them from the manuscript entirely.

Page 24, Line 389: Rewrite conclusions in light of changes to the manuscript.

Page 25, Line 454: This is the incorrect format for this reference. The proper citation format is presented at the beginning of the relevant chapter in the IPCC report. Also, throughout this section, there is no need for the large indent following each new reference. A space between each reference is sufficient. There are a number of other errors throughout the reference section that the Authors should take time to address through careful inspection and consultation with the journal guidelines.

References

Dawson, M. L., Perraud, V., Gomez, A., Arquero, K. D., Ezell, M. J., and Finlayson-Pitts, B. J.: Measurement of gas-phase ammonia and amines in air by collection onto an ion exchange resin and analysis by ion chromatography, Atmos. Meas. Tech., 7, 2733–

2744, doi:10.5194/amt-7-2733-2014, 2014

Erupe, M. E., Liberman-Martin, A., Silva, P. J., Malloy, Q. G. J., Yonis, N., Cocker, D. R., and Purvis-Roberts, K. L.: Determination of methylamines and trimethylamine-N-oxide in particulate matter by non-suppressed ion chromatography, J. Chromatogr. A, 1217, 2070–2073, doi:10.1016/j.chroma.2010.01.066, 2010

Place, B. K., Quilty, A. T., Di Lorenzo, R. A., Ziegler, S. E., and VandenBoer, T. C.: Quantitation of 11 alkylamines in atmospheric samples: separating structural isomers by ion chromatography, Atmos. Meas. Tech., 10, 1061-1078, doi:10.5194/amt-10-1061-2017, 2017

VandenBoer, T. C., Petroff, A., Markovic, M. Z., and Murphy, J. G.: Size distribution of alkyl amines in continental particulate matter and their online detection in the gas and particle phase, Atmos. Chem. Phys., 11, 4319–4332, doi:10.5194/acp-11-4319- 2011, 2011.

VandenBoer, T. C., Markovic, M. Z., Petroff, A., Czar, M. F., Borduas, N., and Murphy, J. G.: Ion chromatographic separation and quantitation of alkyl methylamines and ethylamines in atmospheric gas and particulate matter using preconcentration and suppressed conductivity detection, J. Chromatogr. A, 1252, 74– 83, doi:10.1016/j.chroma.2012.06.062, 2012

Verriele, M., Plaisance, H., Depelchin, L., Benchabane, S., Locoge, N., and Meunier, G.: Determination of 14 amines in air samples using midget impingers sampling followed by analysis with ion chromatography in tandem with mass spectrometry, J. Environ. Monit., 14, 402–408, doi:10.1039/c2em10636a, 2012

---

## Referee Comment (RC3) · Anonymous Referee #3 · 30 Nov 2017

In this study, the authors measured the concentrations of several alkyl amines in the both gas phase and particle phase in boreal forest intermittently over a very long period. Analysis on the temporal variation, possible sources and relationships with meteorological conditions and particle number concentrations were made. This study provides a valuable dataset for the potential source apportionment of amines, which fits the scope of ACP. Highlights are the suggestion that soil can be both sinks and sources. However, in this manuscript, many of the analyses are very vague and are not clearly explained. Some critical information is missing in the manuscript. The authors should consider addressing the following issues before publication on ACP.

Major issues: The introduction to the manuscript consists of one paragraph talking about the importance of amines on new particle formation (NPF), and four paragraphs

introducing existing measurement techniques. However in the results and discussion part, NPF events were not identified, and the detection method was also not the main focus of this paper. The authors should rethink the contents in the introduction so that it can motivate the highlights of this study. Because the authors use a novel measurement technique, it would be valuable if they spent more time explaining its advantages and drawbacks.

As amines are known for their very low ambient concentration (Ge et al., 2011a), it should be mentioned the length of measurement days, the total valid measurement numbers, and number of measurements above detection limits for each amines in each month. When the authors calculate mean or median concentrations, how do they account for the measurements that were below the limit of detection (e.g. in Figures 1 and 2)? Given how frequent these are, it will be very important for the interpretation of their subsequent analyses.

Also, it is hard to understand N numbers in Table 3. For example, DEA has only 6 data above detection limits. However, according to Table 5 and 6, there were at least 81(=79+2) valid gas phase concentration measurement and 26 valid aerosol phase concentration. If there were only 6 measurement with simultaneous detectable level of DEA in both gas phase and particle phase, it means that gas phase was more likely to have detectable concentration considering both channels had the same detection limit (Table 2). In that case, the authors should rethink about the statement made in Line 167 that amines were mainly in aerosol phase. The same problem happens to other amines as well.

In the contents, the authors sometimes miss the indication of the phase in which amines were talking about, such as line 185, line 212, line 230. I suggest the authors use NR3(g), NR3(p), NR3(tot) to indicate gas phase, particle phase and total concentration, respectively.

MARGA measures cations and anions simultaneously. How about anions such as

nitrate and sulfate? They were not mentioned in this study. However, for the study of phase partitioning of amines, it is quite beneficial to learn whether amines are in the form of sulfate salts, nitrate salts or free amines (Ge et al., 2011b).

Line 94 and table 1: The average humidity was very high in March, November and December, was it because of multiple rainy days? How rainfall would affect on-line sampling? Also, indicate the main wind direction.

Line 111: Were particles dried before measurement? If yes, was it before or after the inlet? Also, why chose to collect PM10 instead of PM2.5 or PM1.0?

Line 114: Metrosep C4-100/4.0 is a short column designed for quick measurement of major inorganic ions. Can it separate seven aminiums with no interference from inorganic ions? Does DEA also co-eluent with TMA? It's better for authors to show sample/standard spectrum in the supplement.

Line 115: Where did blank signals of DMA and TMA come from? Was it contamination?

Line 131-133: More clarification.

Line 166 to 168: The data presented in Figure 1 and Table 4 have some discrepancies. The sum of gas phase and particle phase concentration (Table 4) did not equal to the total concentration in Figure 1.

Figure 2: Why no ammonia/ammonium signals in November or December?

Line 188-189: more evidence or discussion is required to draw to that conclusion. Why melting snow could be a source when no linear regression was not identified between air temperature and MMA(g), and even negatively correlated with MMA(p) as stated in Table 5 and 6?

Line 215: show quantitatively about this increase.

Line 222-224: In the study of Dawson et al. (2014), their TMA measured concentration ranged from 1.3-6.8 ppb, not ppm.

Line 230-231: It's hard to tell on the graph when the maximum appeared.

Line 231-232: EA and monoterpene having similar diurnal variation is the main evidence for the authors to address that EA has biogenic source. However, as shown in Figure 5, on July 11th, very high concentration of monoterpenes was observed, while EA concentration remained low. Compared to July 11th, on 12th, the monoterpenes concentration was only half of that on previous day, but EA concentration was more than tripled. On 14th, monoterpene had only one peak while EA exhibited two diurnal peaks. Their behavior was not consistent.

Line 255: The highest mean concentration of amines were usually observed in July, while the maximum concentrations prefer to appear in spring. Were there any intensive sources only in spring?

Line 264-270: Were the diurnal behavior the same for each sampling day? It is hard to tell solely from average data whether they were uniform pattern or influenced by some extreme data. Could DMA come from the re-suspension of soil since the authors measured PM10 (include coarse mode particles)?

According to Figure 6, DMA also had nighttime peak at around 1:00 am. The double peak characteristic of DMA suggested it could be more than light-dependent sources.

Line 296 to 297 and Line 304: $R^2$ is too small to address the linear relationship.

Line 299: Previous text only discussed that MMA could originate from melting snow and ground, not TMA.

Line 344-349: The link between DMA and numbers of 1-2 nm particles is very weak. The authors should consider removing this section. The 'improved' relationship under high RH condition does not support amines contribution to NPF as high RH would suppress NPF (Hamed et al., 2011). Line 378-279: The correlation between PM10 $NH_4^+$ with cluster mode particle numbers is not very meaningful.

Minor issues: Line 27: 0.63?

Line 47: HPLC is the abbreviation for high performance liquid chromatography.

Line 112: It is very unlikely to use 3.2 mol/L oxalic acid as eluent, as oxalic acid solubility under 25 degrees is only 1.6 mol/L.

Line 127: reword.

Line 202: Change ammonia to $NHx=(NH_3+NH_4^+)$

Line 208-209: reword.

Line 215-216: reword.

Figure 3: there are four points largely deviated from the linear regression. Are they included in the calculation of linear regression as well?

Figure 1 and Figure 3: change units to nmol/m3 or neq/m3 when comparing the relative importance of amines with NHx because amines have much higher molecular weight. Put error bars on Figure 1 and Figure 6.

Figure 7: use breaks on x-axis to show clearer time series. Currently, it is hard to tell whether or not DMA shows similar temporal trend as T, ST or SH based on the graph.

Table 6 is not discussed in the main contents, the authors can move it to supplement.

Reference: Dawson, M. L., Perraud, V., Gomez, A., Arquero, K. D., Ezell, M. J., and Finlayson-Pitts, B. J.: Measurement of gas-phase ammonia and amines in air by collection onto an ion exchange resin and analysis by ion chromatography, Atmospheric Measurement Techniques, 7, 2733-2744, 10.5194/amt-7-2733-2014, 2014. Ge, X., Wexler, A. S., and Clegg, S. L.: Atmospheric amines – Part I. A review, Atmospheric Environment, 45, 524-546, 10.1016/j.atmosenv.2010.10.012, 2011a. Ge, X., Wexler, A. S., and Clegg, S. L.: Atmospheric amines – Part II. Thermodynamic properties and gas/particle partitioning, Atmospheric Environment, 45, 561-577, 10.1016/j.atmosenv.2010.10.013, 2011b. Hamed, A., Korhonen, H., Sihto, S.-L., Joutsensaari, J., Järvinen, H., Petäjä, T., Arnold, F., Nieminen, T., Kulmala, M., Smith, J. N., Lehtinen, K. E. J., and Laaksonen, A.: The role of relative humidity in continental new particle formation, Journal of Geophysical Research, 116, 10.1029/2010jd014186, 2011.

---

## Author Response (AR1)

5

30

35

The manuscript "Amines in Boreal Forest Air at SMEAR II Station in Finland" provides an in-situ observation of 7 amine species in both particle and gas phases along with ammonia and ammonium ion over a continental rural area at the Finnish boreal forest site, Hyytiälä, in year 2015 from March to December. The study lasted for a total of 8 weeks, spread out over 8 months. It applies a newly developed measurement technique for amines that combines online ion chromatography and an

10 electrospray ionization quadrupole mass spectrometer. The data analysis relies on simple linear regression to explore the relationships between amines and several environmental factors including rainfall, soil temperature, soil moisture, ambient air temperature and ambient air relative humidity. In addition, the work is used to explore amine species, especially dimethylamine in new particle formation through nucleation process. The authors provide considerate insights of diurnal and seasonal variations of amines over the study region, and highlight the different production mechanisms and sources

**15 among the detected amines.**

The topic of this paper is relevant to the journal and has important scientific contributions to the knowledge of amines in remote continental area, especially with relative longer period observations compared with previous studies. The experimental design is good. However, improvements are needed in the Results section, especially for the quality of figures and tables, in order to deliver to readers more concise and better visualized results. Also, authors should provide more

20 thoughtful interpretations before drawing conclusions. Prior to publication, the authors should address the specific comments below.

1. Please provide detailed information of the sampling period. What was the ratio-

- nale to pick the 8 weeks during the 8 months? Since the study emphasizes seasonal
- 25 variations, how confident can one be with measurements from relative short sampling periods in each month to make conclusions about seasonal changes?

Our plan was to measure from spring to autumn, and not only 8 weeks. Unfortunately, the instrument had failiers and leaks, and 8 weeks was only we could achieve. Our data includes 117 data points in March, 112 in April, 163 in May, 91 in July, 133 in August, 128 in November and 54 in December and we have included this information to the Table 4. This is much more data than has been published earlier.

2. Section 2.2: Authors simply use one sentence to cite previous work as Junninen et al. 2009 without a brief description of what this portal is. A bit more information is warranted. In addition, Junninen et al. 2009 is missing in the reference list. Please check and add in.

In table 1, do environmental conditions have small or big variations during each month? please add standard deviations to each mean value. Also, it would be helpful to make statements of diurnal changes (i.e. day vs. night). Also, please provide information about rain, soil moisture, and soil temperature, as they are important environmental forters in the discussion.

40 factors in the discussion.

"SmartSmear is the data portal for vizualisation and download of continuous atmospheric, flux, soil, tree, physiological and water quality measurements at SMEAR research stations of the University of Helsinki" sentence was added to section 2.2. We also added the missing reference to the list. Table 1 was moved to Supporting Material (Table S1), because Referee 2 asked. We added

45 standard deviations to the Table S1. We also added information about rain, soil humidity and soil temperature to the Table S1, but the day and night means we did not find meaningful to add to the Table.

3. Misleading description at the very beginning of section 3.1: "Figure 1 shows the monthly means and medians of total amine concentrations (sum of gas and aerosol phases) ". Figure 1 only shows means. Correction is needed.

50 It is confusing to claim monthly mean changes as seasonal variations (shown in figure 1) unless the authors define the seasons at first. In the figure, half of the species (EA, DEA, PA and BA) have different scales than the rest. Please consider using two

different y-axis scale in one plot or having two separate plots in order to provide more clear trends for each species.

55 Please clarify the meaning(s)/significance of showing the sum concentrations of gas and particle phases measurements (Figure 1 and 2). Tables 3 and 4 seem to deliver similar cumulative results as Figures 1 and 2 but in separate phases, which are arguably better to understand.

| 60 | The word "median" has been deleted.
We have added which months refer to which season in chapter 2.1.
Sentence "Total amine concentrations were used because we wanted to study amine sources and partitioning between aerosol
and gas phase are dependent on environmental parameters." was added to chapter 3.1.
Figure 1 was moved to Supporting material because the Referee 2 asked.                                                                                                                                                                                                                                                                                                              |
|----|-------------------------------------------------------------------------------------------------------------------------------------------------------------------------------------------------------------------------------------------------------------------------------------------------------------------------------------------------------------------------------------------------------------------------------------------------------------------------------------------------------------------------------------------------------------------------------------------------------------------------------------------------------------------------------------------------------------------|
| 05 | 4. Line 185-190: "The concentration increase in March is characterized with rain (Fig. 4) and the later increase in April took place during night with decreasing wind speed and higher temperature. This increase could be due to evaporation from melting snow and ground." In Figure 4, the time scale on x-axis is too rough to provide ar vision of diurnal variation.                                                                                                                                                                                                                                                                                                                                       |
| 70 | In provement is needed.
It indicates rainfall is featured with high MMA concentration in March (Figure 4), which
is mostly in the particle phase, as shown in Table 4. Does such high MMA relate to
previous cloud processing? Except rainfall, do the other environmental conditions have
potential influences? Authors should expand discussion here.                                                                                                                                                                                                                                                                                                                                               |
| 75 | No detailed information of wind speed and ambient temperature is provided to support the discussion. More explanations and possible references could assist the discussion about evaporation from melting snow and ground.                                                                                                                                                                                                                                                                                                                                                                                                                                                                                        |
| 80 | We have split the Figure 4 in three pieces to make it clearer. We also add wind speed and ambient temperature data in April to the figure.                                                                                                                                                                                                                                                                                                                                                                                                                                                                                                                                                                        |
| 85 |  <li>5. For Section 3.1.4, the authors should include discussion to more extensive literature examining this species. There are a number of references in fact for all of the amines, but for DMA, at the minimum a couple of references with discussion of sources and behavior of DMA are the following:</li> <li>Youn, J. –S., et al. (2015). Dimethylamine as a major alkyl amine species in particles and cloud water: observations in semi-arid and coastal regions, Atmos. Environ., 122, 250-258, doi:10.1016/j.atmosenv.2015.09.061.</li> <li>Murphy, S. M., et al. (2007). Secondary aerosol formation from atmospheric reactions of aliphatic amines, Atmos. Chem. Phys., 7, 2313–2337.</li>  |
| 90 | We have added more discussion to Section 3.2.2 (former 3.1.4) with Youn et al. We could not find much discussion about dimethylamine in Murphy et al, but the reference was valuable to Introduction.                                                                                                                                                                                                                                                                                                                                                                                                                                                                                                             |
| 05 | 6. Figure 5: Time scale on x-axis is too rough to tell diurnal circle. Improvement is needed.                                                                                                                                                                                                                                                                                                                                                                                                                                                                                                                                                                                                                     |
| 95 | More detailed time scale for x-axis was added.                                                                                                                                                                                                                                                                                                                                                                                                                                                                                                                                                                                                                                                                    |
| 00 | 7. Section 3.1.4: Authors emphasize DMA (and TMA) concentration higher in summer
(i.e. August) due to biogenic sources. However, interpretations/discussion leading
to that conclusion are not convincing in my opinion. It mentioned that DMA does not                                                                                                                                                                                                                                                                                                                                                                                                                                                     |

- 100 to that conclusion are not convincing in my opinion. It mentioned that DMA does not show correlation with biogenic tracer such as monoterpenes, while isoprene is noted as having light dependent emissions. Please provide supportive BVOC tracer information if the data is applicable. Are there BVOC tracers other than isoprene found related to variations of DMA and (or) TMA?
- 105 In Figure 6, DMA shows strong diurnal cycle while TMA doesn't. Is the DMA diurnal circle found only during summer, especially August? Why is it that TMA does not have such a strong diurnal circle as it also mentioned in section 3.1.2? Authors should expand discussions here.

110

The DMA diurnal cycle is found only in summer. It is determined by the balance between emissions, reactivity and mixing in the atmosphere. Therefore the compounds emitting from the same sources can have different atmospheric concentrations. Usually the diurnal variation is mainly determined by mixing, causing daytime minima, but if emissions are light dependent or strongly temperature dependent, then the maxima is at daytime. This is mentioned in section 3.2.4. However, we have not mentioned in the

115 text that higher summer time concentrations also indicate biogenic sources. This has been added to the text. Sources and source areas of DMA and TMA are not known, and different diurnal cycles can be caused by different balance between emissions, reactivity and mixing.

8. Figure 7: The current plot is hard to show the clear relationship between DMA and selected environmental factors, especially for data around July and December. Please consider zooming in time scale on x-axis, for an example, using a discontinued time series.

We have made the Figure clearer according to the reviewers wish.

```
125
```

9. Line 295- 300: Does TMA negatively correlate with ambient temperature consistently or is this sensitive to season?

130 Yes, it is sensitive to season, we added a picture to Supplement (Figure S5).

**Minor comments**

135 1. Authors should consider adding a site map in section 2.1 in order to provide readers visualized information of study area.

We have added a site map to Supporting material.

140 2. Line 115: Why are the DL calculation methods for DMA and TMA different from the rest? In table 2, DMA shows different DLs in two time periods, while TMA doesn't. Please clarify the reason(s).

The DMA and TMA DLs were calculated from blank-values, because they had some blank, when the other compounds did not. We have added explanation to the text.

3. Table 1: What is the difference between "mean" and "average"? If they are same, please be consistent.

Grey shade is not necessary if the color does not have meaning. Same comment applied to Table2.

We have changed "averages" to "means" and took the color off.

155 4. Table 2: Some species have a comma after their names while the others do not. Why NH3 and NH4+ are not mentioned in gas (particle) phase as the rest? Keep consistent style please.

We have added commas after every amine. We have added "gas" and "particle" after ammonia and ammonium. 160

5. Table 4: Typo for DMA median value (particle phase) in July. "4,9" should be "4.9".

We have changed it from "4,9" to "4.9".

165 6. Figure 4: Add label for x axis. Same comment for Figure 5. The x-axis represents dates, but it is unclear. In contrast, Figure 7 has better x axis format. ase be consistent in plot style.

We have added the labels.

7. Figure 7: Units on y axis should be in parentheses.

**175 Interactive comment on **"Amines in Boreal Forest** Air at SMEAR II Station in Finland" bv Marja 180 Hemmilä et al.**

Anonymous Referee #2 Received and published: 16 November 2017

**General Comments**

185

The Authors present several weeks of ambient observations of seven atmospheric alkyl amines in the gas and particle phases using the MARGA system interfaced with a quadrupole mass spectrometer. The quantities of these amines from each of the observation periods is reported and trends discussed from a variety of perspectives (e.g.

190 diurnal, inter-monthly, etc.) They investigate a variety of correlations of the measured amines with accompanying physical observations made at the same location. Overall, this manuscript is not suited for publication in Atmospheric Chemistry and Physics without many major issues being addressed. **Major Comments**

195

1. There are far too many data tables and figures that are irrelevant to the structure of the results and discussion. Most can be replaced with a single sentence. These detract from the quality of the observations and should be relocated to a Supporting 200 Information document. These are noted in detail below.

Asked figures and tables were moved to Supporting Material.

2. The Authors claim that they are reporting the longest time series of amines mea-205 surements to date, but the measurements are short-duration periods made in different months. A time series implies continuous data collection and the authors should revise the manuscript to be clear that they are reporting eight weeks of observations from different months over the course of a year from the same observation site. Further to this point, the sampling strategy reported and the findings discussed in the paper are

210 all weakened because of the intermittent nature of these observations. The limitations of the dataset need to be presented clearly. Figure 7 is the only depiction of the full measurement time series and it appears that even within each observation period that there are gaps in the data which are not clearly explained. How can the Authors justify their conclusions regarding monthly/seasonal trends if they do not measure continu-

215 ously throughout each?

> Number of data points in each month was added to Table 4. We also added clarification in Experimental section that due to instrumental problems good quality data was captured only 8 weeks, although we measured continuously. Even though measurements cover only 8 weeks, to our best knowledge, this is still largest data set of amine concentrations.

220

3. The manuscript does not appear to have a clear purpose or objective. There are several discussions made throughout the manuscript that are not joined in a clear narrative, the final paragraph of the introduction for example, which confuses the meaning and scientific contribution of the findings.

a. The Authors present an advancement in atmospheric amine measurement capabilities through the use of an ion chromatography-mass spectrometry system, but do not clearly demonstrate the necessary performance metrics (e.g. a sample chromatogram demonstrating the ability to speciate the suite of analytes likely to be accountered in

- 230 demonstrating the ability to speciate the suite of analytes likely to be encountered in the atmosphere). The data then presented in the tables and figures is still largely below the instrument detection limits (e.g. Table 4), so is this improvement really meeting the observational needs of the research community?
- 235 We have added chromatogram to Supplement Material. It would be great to have better time resolution and lower detection limits, but this is the best which is possible at the moment. Direct mass spectrometric methods have lower detection limits and higher time resolution, but data is not species specific.

b. A timeseries of diurnal observations and linkages to known boreal biogenic pro-240 cesses is discussed for some amines.

We added this sentence to Introduction.

c. The relationship of gaseous DMA to a variety physical observations is made, con-

- 245 cluding that new particle formation is higher when both DMA and RH are high. To me, this manuscript is a first long-term survey of sources and phase-distribution of amines at the sub-pptv level in a remote boreal forest environment using supporting physical measurements to initiate a better understanding of this entire class of compounds relative to what we already know about ammonia. The Authors should
- 250 carefully review the findings of their work and convey the purpose of their work clearly as it will also strengthen the structure of their results and discussion. The introduction should be subject to a major revision based on the determined purpose and presented structure of the resulting manuscript as it currently does not do so.
- 255 Introduction was improved.

4. The methods section is not detailed enough to evaluate whether the DLs for the MARGA-MS are robust and reliable. There are many issues here that need to be addressed that are detailed in the Technical Comments below.

260

265

Method section was corrected according to Technical Comments

5. The manuscript has many typographical and technical errors that should have been addressed prior to submission (e.g. use of the term 'aerosol particles'). The Authors are strongly encouraged to seek external review of their work by peers and colleagues after revision prior to resubmission for further review.

The manuscript has been checked by a native speaker.

270 Technical Comments

The following comments are not an exhaustive list of the corrections required for this manuscript to be acceptable for publication. The Authors, in addressing the major comments, will likely correct much of the unmentioned issues under their own revision.

275

Page 1, Line 15: The Authors state that they can separate and detect 7 different amines, but do not show any evidence of this performance, nor does the literature cited for the methodology used. If there is prior work demonstrating the quality of this method performance for amines, specifically, it should be cited and briefly summarized.

280 If not, then the Authors are missing an opportunity to present a significant advance in the online measurement of atmospheric amines simultaneously in the gas and particle phases. The chromatograms have been added to Supporting Material to demonstrate the separation of amines, and we have added more information about the method to the Experimental and Results sections.

Page 1, Line 17: The term 'possibly' is used here, which is speculative. Such conclusions, while potentially acceptable for the discussion, should not be present in an Abstract. Major findings with solid support only should be presented here.

290

The sentence with 'possibly' was deleted.

**Page 1, Lines 21-22: There is no conclusive evidence presented for DMA and TMA biogenic sources. This is only true for EA.**

295

EA correlated with monoterpenes, which has temperature dependent sources. DMA and TMA may for example have light dependent sources, like isoprene, or they may be emitted from soil.

**Page 1, Lines 23-24: What is the purpose of presenting the means and medians for 300 these measurements in the Abstract? This does not seem necessary.**

In our opinion the concentration levels are the most important and newest result, and therefore we want to include them in the abstract.

305 Page 1, Line 27: '0.63 EA' This is an example of missed typographical errors that should be identified prior to manuscript submission.

We corrected the miss-typings.

310 Page 2, Line 34: The amine class of compounds can be more broadly defined. R- can be used to represent both H- and alkyl- substituents.

We wanted to make a clear difference between ammonia and amines.

315 Page 2, Lines 35 and 38: 'aerosol particle' is incorrect terminology. These are interchangeable terms and typically one or the other is chosen for use throughout a manuscript.

We changed so that we use only 'aerosol', when it is possible.

320

Page 2, Lines 44-45: 'and' is used twice in a row. Another example of missed typographical error.

We corrected that.

325

Page 3, Lines 67-68: The Authors have not presented any information on the range of amines that have been measured, or are even estimated, to be present in the boreal forest prior to this sentence evaluating the potential utility of prior published methods. Depending on the chosen purpose of the manuscript, the Authors should either ex-

330 pand on the findings of previous boreal measurements, focus on the performance of the measurement technique relative to other reports, or link these two themes with a motivation of greater breadth.

We took off the sentence with DLs, and replaced it with "in this method ammonia/ ammonium samples could impede detection of some amines". However, the method was guite impressing.

Page 3, Line 89: Why are the exact dates for the measurements not given? The datasets, such as Figure 7, clearly show that these periods were not subject to continuous observation and the details should be provided here.

340

Measurements were continuous, except when we were calibrating, measuring blank or cleaning the instrument, or when the instrument was broken. Unfortunately, even than we visited the instrument weekly and checked it via internet almost daily, and

somebody in the field visited it almost daily, still sometimes afterwards we noticed, that something went wrong; so we took that data off. Number of proper measurement data points during each month was added to Table 4.

345

Page 4, Table 1: Move this to the SI and add soil humidity and soil temperature details to this section of the methods since they don't appear anywhere, but appear regularly throughout the results and discussion.

350 We have moved Table 1 to the Supplement Material.

Page 4, Line 103: Define the MARGA acronym first and then put the acronym in brackets.

355 We have changed that.

Page 4, Lines 105-106: This is far too brief of a description for the interfacing and operation of a mass spectrometer to an ion chromatograph. Does the MARGA suppress the IC eluent prior to analyte measurement? Were conductivity measurements made

360 prior to, but in series, with the mass spectrometer? What solvent and what ratio to the IC mobile phase was added prior to the electrospray? What were the desolvation and transmission settings of the mass spectrometer?

We have added the information to the text. MARGA does not suppress the IC eluent in cation side. Waste line from cation conductivity detector was leaded to ESI-needle, and no additional solvent was added. The table about MS settings was added to Supplement Material.

Page 4, Lines 112-113: Was the separation an isocratic or gradient separation? What was the time required for separation? What were the dimensions of the analytical sectors and the transfer of the sectors are set of the sectors and the sectors are set of the sectors and the sectors are set of the sectors and the sectors are set of the sectors

370 column and what was the particle size of the stationary phase?

We have added the information to the text. Separation was isocratic, and the time was 15 min.

- 375 Page 4, Lines 114-116: Are these the DLs for the IC system or for the MARGA? That is, were the S/N = 3 calculated from blank injections, the measurements made in 'blank-mode', or from MARGA solvents analyzed when the inlet was overflowed with zero air? Given the need for very sensitive measurement capabilities outlined in the introduction, the Authors should present a more detailed description of how the DLs
- 380 were determined and the quantitative metrics evaluated to conclude that they are reliable and robust representations of the capabilities of the MARGA. Characterizations of these parameters for IC systems measuring amines have been previously described by Erupe et al. (2010), Dawson et al. (2014) and Place et al. (2017), those with atmospheric interfaces by VandenBoer et al. (2011, 2012), and IC coupled to mass
- 385 spectrometry by Verriele et al. (2012)

The DLs are for the whole MARGA-MS –system (excluding the inlet), and the measurements were made in blank-mode. The references were gratefully checked and most of them were cited in the manuscript.

- 390 Page 5, Table 2: This belongs in the results section of the manuscript along with the remainder of the analytical performance metrics. The Authors could consider reporting the DLs as ng/m3 for the particle channel and pptv for the gas channel to improve the clarity of this table.
- 395 We have moved Table 2 to results section.

Page 5, Lines 122-125: How was the d10-DEA introduced to the MARGA-MS as an internal standard? Was it added to the solvent in the particle and gas channels, to the post-suppressor organic solvent, or only to standards that were injected offline?

400 What is the purpose of using this internal standard and how did it perform for the set of reported field measurements? Presumably it was used to track the spray ionization efficiency on an ongoing basis in the post-suppressor organic solvent, but this is not clear. The Authors should also indicate how the internal standard was utilized in back calculating the quantities of all the measured amines and whether it reduced uncertainty in the measurements and by how much.

405 the measurements and by how much. Also, why is are the calibration standard values listed as '~' which means approximate? Further to this, the calibration range seems to be far above the observed values, which are typically below 10 ng/m'3, which is the lowest calibration standard, and is therefore extrapolating the calibration below the determined range. The Authors should

- 410 provide proof that the system sensitivity response from 0.1-10 ng/m3 is the same as from 10-300 with at least a 3-point calibration for all of the analytes. They should use this information to determine their measurement accuracy and precision as well, which is not evaluated.
- 415 We have added to the text "DEA10 was used, because it behaved same way in IC-separation but had different mass than studied amines. 50,0 µl of DEA10 was added to MARGAS ISTD solution bottle. When MARGA was taking the air sample, it was at the same time taking ISTD solution to similar sample syringes. When IC-analysis started, the ISTD and air sample solution mixed, and this solution went to IC-separation, conductivity detection and finally MS-detection. DEA10 was used to correct possible loses to instrumentation and correct changes of MS response." ISTD went through the same analysis than analytes, so it corrects the
- 420 possible biases of the process. We have added accuracy and precision to Table 1.

Page 5, Lines 128-129: Since the MARGA was not operated continuously for the analysis of amines, it is important to specify whether the instrument blank values were collected before, during, or after the periods when continuous monitoring was being

425 performed. Again, I presume this was performed immediately before, intermittently during, and immediately after each observation period so that backgrounds could be corrected throughout each observation period, but the explicit information needs to be presented in the manuscript.

430 Because MARGA-MS was running all the time (except when it was broken, or we were calibrating or cleaning it), the blank values were supposed to collect once a month. Unfortunately the instrument had habit to stop just before or during the blank running, so we missed those blanks.

Page 5, Lines 131-133: If the DMA backgrounds are different due to cycling between sampling syringes, it should be possible to explicitly assign a background correction to samples collected with each set of syringes instead of averaging, which will decrease both accuracy and precision of the measurement.

We were able to get better results with averaging, since blank subtraction did not correct the difference totally.

440

Page 6, Line 135: This should be the start of a new methods section that describes the DMPS measurements.

Corrected.

- 445
  - Page 6, Line 149: Regression calculations can allow some insight into the physicochemical nature of the amines, but the results and discussion do little to explore the reasons why the variable the authors chose were investigated. What are the chemical and physical mechanisms that may be acting to release amines in the boreal environ-
- 450 ment? Is there a precedent from laboratory work or prior observations? Given those parameters explored, the working hypothesis seems to be testing whether there is similar release and exchange of amines in the boreal as might be expected given the extensive literature on ammonia. If this is the case, this reasoning should be emphasized throughout the manuscript and supported by citing the relevant literature.
- 455

The chemical and physical mechanisms behind the emissions are beyond the scope of this manuscript.

Page 7, Lines 167-168: 'amines were mainly in the aerosol phase' - the averages appear very close to 0.5 and likely are if the variability in the data is considered. The

460 range of the gas fraction values presented in Table 3 suggests this is the case. It

**would be more accurate to say that the gas fraction was variable, with an average and standard deviation given for the entire dataset.**

We included number of gas phase data and number of aerosol phase data above detection limit, and with this information it is correct to say, that amines were most in the aerosol phase.

Page 7, Figure 1: Move this to the Supporting Information. It repeats all of the data presented in Figure 2 and since the observations do not span the entire month, it may be more accurate to add the observation dates to the bar labels in the legend. It would

470 be more informative for the authors to present some continuous data in a figure that includes many of amines, ammonia/ammonium, the uncertainty in each measurement, and the DLs so that the quality of the MARGA-MS data can be ascertained.

We have moved the Figure 1 to the Supporting Material

475

Page 8, Figure 2: How many points are off scale in each of these panels? Why are the DEA measurements suggesting that there were negative quantities detected? Where are the November and December measurements for NH3+NH4?

480 Off scale bars showed the maxima of all measurements and this value is also shown in the figure. There are no negative values for DEA, minima are just detection limits. For some reason ammonia and ammonium data in winter was forgotten, but we have now added it.

**Page 8, Table 3: Please check that the table formatting is done according to the guide-**

485 lines of ACP

We have changed the form of Table 3 (and other tables too).

Page 9, Table 4: Move this table to the Supporting Information and replace with 2-3 simple sentences in the text. Why are both the mean and median values in this table? What does a difference between these two metrics tell us about the amine measurements and why? This table suggests that even MARGA-MS does not have adequate DLs for the boreal environment and the authors should comment on this in the discussion.

495

We have moved the Table 4 to the Supporting Material. With mean values we showed, that even though median (i.e. most of the data) was <DL, in some cases there still was remarkable concentrations above DL.

500 Page 10, Figure 3: Remove '(y-axis)' and '(x-axis)' from the caption. This is obvious. Consider a more descriptive caption.

We have removed '(y axis)'and '(x-axis)'.

- 505 Page 10, Figure 4: This date format is not consistent with previous figures, and the notation is not defined via the axis label. The data presented in this figure are clearly non-continuous between months and goes back to the points above regarding statements by the Authors suggesting that the dataset is continuous when it is not. What is the purpose of plotting rainfall on this figure? The discussion speculates on
- 510 'evaporation from melting snow and ground', but rainfall does not describe either of these processes. What is the physical or chemical rationale for this speculation? Is there precedent in the literature to support this?

We have added now the Figure 4b, which shows the effect of high night time temperature and decreasing wind speed, and this is also discussed in the text.

Pages 11 -18: The Authors compare their measurements for each species to those from other reports throughout. This would be more easily conveyed through the use of tables that present the data from this work in comparison to the findings of others,

**520 listing relevant parameters such as rural and urban settings.**

We have made a table, that compiles the results of our and other studies.

Page 11, Line 210: Why is TMA discussed before EA, but presented in Figure 6 follow-525 ing the description of EA in Figure 5? Please reorganize the discussion so that figures appear in the order that they are discussed.

We have re-organized the sections

- 530 Page 12. Figure 5: Most of the observations for EA are below the DL, so why is the line in the figure continuous? It would be more appropriate to add another trace here that denotes the cut off for the DL and to leave gaps in the EA dataset where the measurement is below the DL.
- 535 We changed the second x-axis so, that EA concentrations below DL are under it.

Page 13, Lines 265-266: Soils and surfaces in the boreal are acidic when measured in bulk, so it seems unlikely that deposition and re-emission is a plausible line of speculation. For example, a comparison using compensation point theory for ammonia in

540 these environments could suggest that deposition should be the final fate of bases at the surface, and could support a similar case for the amines since they are stronger bases than ammonia. This would be strong evidence against deposition and desorption cycles and indicate other mechanisms of emission (e.g. decomposition of organic matter).

545

The surfaces of leaves and needles are probably not that acidic, but it is not known based on the discussion with forest scientists.

Page 13, Figure 6: Why did the Authors not explore the diurnal nature of TMA in

- 550 July when the measured mixing ratios were highest? This compound is known to be released during the flowering of many plants to attract pollinators and should be discussed for context.
- We chose to show diurnal variation in August, because it was more pronounced for DMA then. For TMA July and August looked the 555 same.

Page 14, Line 279: 'concentrations of DMA vary with temperature' - Figure 7 does not demonstrate such a dependence, but the statistical findings in Table 5 do. There is no clear dependence to the eye in Figure 7 between DMA and air temperature.

560

We have made the Fig. 7 clearer, and also referenced to Table 5.

Page 14, Figure 7: These panels are not alphabetically labeled for reference in the

caption. Previous figures suggest there were DMA measurements made in August, but they are not shown on this figure. Why is this? The findings in Table 5 suggest that the correlations between DMA and a number of parameters are worth 565 noting and those plots would be more valuable than this figure. Consider replacing Figure 7 with a multi-panel figure showing these relationships and the regression statistics from Table 5.

We have checked that all the DMA(g) data were <DL in August. We added scatter plots between DMA(g) and ambient parameters shown in Fig. 7 in to Fig. 8 according to reviewers suggestion. However we wanted to keep time series as well, because seasonal behavior is more obvious there.

570

Page 15, Table 5: There is very little here that is meaningful to the discussion. It can be replaced with the figure noted in the previous comment and a couple sentences in the text. Move to the SI or consider removing from the manuscript entirely.

Table 5 was moved to Supplement. We wanted to show that most of the compounds are not depending on ambient conditions.

**Page 17, Table 6: Same comment as Table 5.**

580 Table 6 was moved to Supplement.

Page 19-22, Figures 8 and 9, Tables 7 and 8: Only depict and present the most meaningful results. That is, present the figure panels that convey information central to the discussion which cannot be easily replaced with three or fewer sentences in the text.

585 In this case, consider combining the most important findings from Figures 8 and 9 into one figure. Move the tables to the Supporting Information or consider removing them from the manuscript entirely.

We have moved the Tables in to the Supplement. We removed Fig. 9, since the particles in aerosol phase observed by MARGA-MS cannot explain the cluster mode particle number concentration.

Page 24, Line 389: Rewrite conclusions in light of changes to the manuscript.

We have done that. 595

Page 25, Line 454: This is the incorrect format for this reference. The proper citation format is presented at the beginning of the relevant chapter in the IPCC report. Also, throughout this section, there is no need for the large indent following each new reference. A space between each reference is sufficient. There are a number of other

600 errors throughout the reference section that the Authors should take time to address through careful inspection and consultation with the journal guidelines.

We have checked the reference lists and corrected the miss-typings.

630

**635 Received and published: 30 November 2017 In this study, the authors measured the concentrations of several alkyl amines in the both gas phase and particle phase in boreal forest intermittently over a very long period. Analysis on the temporal variation, possible sources and relationships with meteorological conditions and particle number concentrations were made. This study pro vides a valuable dataset for the potential source apportionment of amines, which fits the scope of ACP. Highlights are the suggestion that soil can be both sinks and sources. However, in this manuscript, many of the analyses are very vague and are not clearly explained. Some critical information is missing in the manuscript. The authors should consider addressing the following issues before publication on ACP.**

**645**

Major issues: The introduction to the manuscript consists of one paragraph talking about the importance of amines on new particle formation (NPF), and four paragraphs introducing existing measurement techniques. However in the results and discussion

650 part, NPF events were not identified, and the detection method was also not the main focus of this paper. The authors should rethink the contents in the introduction so that it can motivate the highlights of this study. Because the authors use a novel measurement technique, it would be valuable if they spent more time explaining its advantages and drawbacks.

**655**

685

We have improved the Introduction. We also have added more details and evaluation of the method to Experimental and Results sections

As amines are known for their very low ambient concentration (Ge et al., 2011a), it should be mentioned the length of measurement days, the total valid measurement numbers, and number of measurements above detection limits for each amines in each month. When the authors calculate mean or median concentrations, how do they account for the measurements that were below the limit of detection (e.g. in Figures 1 and 2)? Given how frequent these are, it will be very important for the interpretation of

665 their subsequent analyses.

More detailed descriptions about the measurements have been added. In Table S2, number of data points in each month is presented. When we are calculating means or medians, the values bolew DL were taking account as 0,5\*DL.

- 670 Also, it is hard to understand N numbers in Table 3. For example, DEA has only 6 data above detection limits. However, according to Table 5 and 6, there were at least 81(=79+2) valid gas phase concentration measurement and 26 valid aerosol phase concentration. If there were only 6 measurement with simultaneous detectable level of DEA in both gas phase and particle phase, it means that gas phase was more likely to
- 675 have detectable concentration considering both channels had the same detection limit (Table 2). In that case, the authors should rethink about the statement made in Line 167 that amines were mainly in aerosol phase. The same problem happens to other amines as well.
- 680 We have improved former Table 3 (now Table 2). Beforehand it only presented the data above DL at the same time in gas and aerosol phase.

In the contents, the authors sometimes miss the indication of the phase in which amines were talking about, such as line 185, line 212, line 230. I suggest the authors use NR3(g), NR3(p), NR3(tot) to indicate gas phase, particle phase and total

concentration, respectively.

We have changed to NR3(g), NR3(a) and NR3(tot).

- 690 MARGA measures cations and anions simultaneously. How about anions such as nitrate and sulfate? They were not mentioned in this study. However, for the study of phase partitioning of amines, it is quite beneficial to learn whether amines are in the form of sulfate salts, nitrate salts or free amines (Ge et al., 2011b).
- 695 Unfortunately we don't have the data from anion side.

Line 94 and table 1: The average humidity was very high in March, November and December, was it because of multiple rainy days? How rainfall would affect on-line sampling? Also, indicate the main wind direction.

700

There was rain and also in March melting snow and ground. Inlet line was sheltered for rain. We have added the main wind direction to Table S1.

Line 111: Were particles dried before measurement? If yes, was it before or after the inlet? Also, why chose to collect PM10 instead of PM2.5 or PM1.0?

No, the particles were not dried, because in Steam Jet Aerosol Collector they get wet, and also the eluent is water. We chose PM10 because it was available and commonly used with MARGA.

- 710 Line 114: Metrosep C4-100/4.0 is a short column designed for quick measurement of major inorganic ions. Can it separate seven aminiums with no interference from inorganic ions? Does DEA also co-eluent with TMA? It's better for authors to show sample/standard spectrum in the supplement.
- 715 We have added the chromatogram of standard to Supplement Material. DEA and TMA were co-eluating a bit, but MS detection separate them totally due to different masses.

Line 115: Where did blank signals of DMA and TMA come from? Was it contamination?

720 It was instrument background.

**Line 131-133: More clarification.**

- We added to the text that we were not able to found more accurate reason for that.
- 725

Line 166 to 168: The data presented in Figure 1 and Table 4 have some discrepancies. The sum of gas phase and particle phase concentration (Table 4) did not equal to the total concentration in Figure 1.

730 Former Fig. 1 (now Fig. S4) contains also the values below the detection limit as 0.5\*DL. In the former Table 4 (now Table S2) they are marked as <DL.

**Figure 2: Why no ammonia/ammonium signals in November or December?**

735 The signals were added, for some reason they were forgotten.

Line 188-189: more evidence or discussion is required to draw to that conclusion. Why melting snow could be a source when no linear regression was not identified between air temperature and MMA(g), and even negatively correlated with MMA(p) as stated in Table 5, and 60.

740 Table 5 and 6?

Also mixing and reactivity affect the concentrations of amines, and therefore hourly values do not correlate directly with temperature. In Table 5 and 6 there are the data from whole year, and snow melting period is not studied independently.

**745 Line 215: show quantitatively about this increase.**

Since this statement was too weak, we took of the sentence.

- Line 222-224: In the study of Dawson et al. (2014), their TMA measured concentration 750
  - ranged from 1.3-6.8 ppb, not ppm.

Corrected

- Line 230-231: It's hard to tell on the graph when the maximum appeared.
- 755

Figure was clarified.

Line 231-232: EA and monoterpene having similar diurnal variation is the main evidence for the authors to address that EA has biogenic source. However, as shown in

760 Figure 5, on July 11th, very high concentration of monoterpenes was observed, while EA concentration remained low. Compared to July 11th, on 12th, the monoterpenes concentration was only half of that on previous day, but EA concentration was more than tripled. On 14th, monoterpene had only one peak while EA exhibited two diurnal peaks. Their behavior was not consistent.

**765**

Concentrations in ambient air are determined by the balance between emissions, reactivity and mixing in the atmosphere. We are not claiming that sources are exactly the same, but similar. The source areas may not be the same either, and therefore wind direction affects too.

770

Line 255: The highest mean concentration of amines were usually observed in July, while the maximum concentrations prefer to appear in spring. Were there any intensive sources only in spring?

775 We are thinking melting snow, as we say in the text. We are going to study the spring snow more in future.

Line 264-270: Were the diurnal behavior the same for each sampling day? It is hard to tell solely from average data whether they were uniform pattern or influenced by some extreme data. Could DMA come from the re-suspension of soil since the authors

780 measured PM10 (include coarse mode particles)?

Diurnal variation for every measurement day (tot. 5) were similar. We expect that amines are in small particles.

**According to Figure 6, DMA also had nighttime peak at around 1:00 am. The double**

785 peak characteristic of DMA suggested it could be more than light-dependent sources.

This is true and we have added a sentence to the text.

**Line 296 to 297 and Line 304: R2 is too small to address the linear relationship.**

790

We agree the reviewer, however, looking at the summer data only, there is a positive correlation between temperature and TMA(g). We added a figure to Supplement (Fig. S6).

Line 299: Previous text only discussed that MMA could originate from melting snow

795 and ground, not TMA.

We added discussion also to chapter 3.2.2.

Line 344-349: The link between DMA and numbers of 1-2 nm particles is very weak. 800 The authors should consider removing this section. The 'improved' relationship under high RH condition does not support amines contribution to NPF as high RH would suppress NPF (Hamed et al., 2011).

We considered it is important to show it is weak, since there has been lots of discussion about the contribution of DMA to NPF. We 805 saw contradictory result than Hamed et al., and that is important to show.

**Line 378-279: The correlation between PM10 NH4+ with cluster mode particle numbers is not very meaningful.**

We agree, but that is an important information too. The Table 8 was moved to Supplement.  $810\,$

Minor issues: Line 27: 0.63?

We took that off.

815

Line 47: HPLC is the abbreviation for high performance liquid chromatography.

We have changed that

820 Line 112: It is very unlikely to use 3.2 mol/L oxalic acid as eluent, as oxalic acid solubility under 25 degrees is only 1.6 mol/L.

There was a typing mistake; we have changed the unit to mmol/l.

**825 Line 127: reword.**

We did.

**Line 202: Change ammonia to NHx=(NH3+NH4+)**

830 We did.

Line 208-209: reword.

835 We did.

840

845

```
Line 215-216: reword.
```

We did.

Figure 3: there are four points largely deviated from the linear regression. Are they included in the calculation of linear regression as well?

Yes they are.

Figure 1 and Figure 3: change units to nmol/m3 or neq/m3 when comparing the relative importance of amines with NHx because amines have much higher molecular weight.

850

865

Put error bars on Figure 1 and Figure 6.

855 Figure 7: use breaks on x-axis to show clearer time series. Currently, it is hard to tell whether or not DMA shows similar temporal trend as T, ST or SH based on the graph.

We did

860  $\,$  Table 6 is not discussed in the main contents, the authors can move it to supplement. We did.

Reference: Dawson, M. L., Perraud, V., Gomez, A., Arquero, K. D., Ezell, M. J., and Finlayson-Pitts, B. J.: Measurement of gas-phase ammonia and amines in air by collection onto an ion exchange resin and analysis by ion chromatography, Atmospheric Mea-

| 870 | Ge, X., Wexler,
A. S., and Clegg, S. L.: Atmospheric amines – Part I. A review, Atmospheric Environ-
ment, 45, 524-546, 10.1016/j.atmosenv.2010.10.012, 2011a.                                                                                                 |                            |
|-----|----------------------------------------------------------------------------------------------------------------------------------------------------------------------------------------------------------------------------------------------------------------------|----------------------------|
| 070 | Ge, X., Wexler, A. S., and
Clegg, S. L.: Atmospheric amines – Part II. Thermodynamic properties and gas/particle
partitioning, Atmospheric Environment, 45, 561-577, 10.1016/j.atmosenv.2010.10.013,
2011b.                                                 |                            |
| 875 | Hamed, A., Korhonen, H., Sihto, SL., Joutsensaari, J., Järvinen, H., Petäjä,
T., Arnold, F., Nieminen, T., Kulmala, M., Smith, J. N., Lehtinen, K. E. J., and Laakso-
nen, A.: The role of relative humidity in continental new particle formation, Journal of |                            |
| 880 | Geophysical Research, 116, 10.1029/2010jd014166, 2011.                                                                                                                                                                                                               |
|     |                                                                                                                                                                                                                                                                      |                            |
| 885 |                                                                                                                                                                                                                                                                      |                            |
| 890 |                                                                                                                                                                                                                                                                      |                            |
| 895 |                                                                                                                                                                                                                                                                      |                            |
| 900 |                                                                                                                                                                                                                                                                      |                            |

**Amines in Boreal Forest Air at SMEAR II Station in Finland**

905 Marja Hemmilä1, Heidi Hellén1, Aki Virkkula1,2, Ulla Makkonen1, Arnaud P. Praplan1, Jenni Kontkanen2, Lauri Ahonen2, Marku Kulmala2, Hannele Hakola1

1 Finnish Meteorological Institute, P.O. Box 503, FI-00101 Helsinki, Finland

2 Department of PhysicsInstitute for Atmospheric and Earth System Research/Physics, Faculty of Science, University of Helsinki, P.O. Box 64, 00014 Helsinki, Finland

Correspondence to: Marja Hemmilä (marja.hemmila@fmi.fi)

- 915 Abstract. We measured amines in boreal forest air in Finland both in gas and particle phase with 1-hour time resolution using an online ion chromatograph (instrument for Measuring AeRosols and Gases in Ambient Air, MARGA) connected to an electrospray ionization quadrupole mass spectrometer (MS). The developed MARGA-MS method was able to separate and detect 7 different amines: monomethylamine (MMA), dimethylamine (DMA), trimethylamine (TMA), ethylamine (EA), diethylamine (DEA), propylamine (PA) and butylamine (BA). The detection limits of the method for amines were low (0.2–
- 3.1 ng m3), the accuracy of IC-MS analysis was 11–37% and the precision 10–15%. The proper measurements in the boreal forest covered about 8 weeks between March 2015 and December 2015. With MARGA MS we were able to separate and detect 7 different amines: monomethylamine (MMA), dimethylamine (DMA), trimethylamine (TMA), ethylamine (EA), diethylamine (DEA), propylamine (PA) and butylamine (BA). The amines were found to be an inhomogeneous group of compounds, showing different seasonal and diurnal variability. Total MMA (MMA(tot)) peaked together with the sum of
- 925 ammonia and ammonium ion already in March, possibly due to evaporation from melting snow and ground. In March monthly means for MMA were <2.4 ng m-3 and 6.8±9.1 ng m-3 in gas and aerosol phase, respectively, and for NH3 and NH4+, 52±16 ng m-3 and 425±371 ng m-3, respectively. Monthly medians in March for MMA(tot), NH3 and NH4+, were <2.4 ng m-3, 19 ng m-3 and 90 ng m-3 respectively. DMA(tot) and TMA(tot) had summer maxima indicating biogenic sources. We observed diurnal variation for DMA(tot) but not for TMA(tot). The highest concentrations of these compounds
- were measured in July. ThenIn July monthly means for DMA were <3.1 ng m-3 and 8.4±3.1 ng m-3 in gas and aerosol phase, respectively, and for TMA 0.4±0.1 ng m-3 and 1.8±0.5 ng m-3. Monthly medians in July for DMA were <DL and 4.9 ng m-3 in gas and aerosol phase, respectively, and for TMA 0.4 ng m-3 and 1.4 ng m-3. When relative humidity of air was >90%, gas phase DMA correlated well with 1.1\_-2 nm particle number concentration (R2=0.63) suggesting that it participates in atmospheric clustering<del>new particle formation. 0.63</del> EA concentrations were low all the time. Its7 July means were <0.36 ng</li>
- 935 m-3 and 0.4±0.4 ng m-3 in gas and aerosol phase respectively, but they-individual concentration data correlated well with monoterpene concentrations in July. Monthly means of PA and BA were all the time below detection limits.

**940 1. Introduction**

In atmospheric chemistry and secondary-aerosol production bases are crucial since they can neutralize acids and therefore accelerate several processes, like e.g. subsequent growth of newly born aerosol particles. Furthermore bases are significant since they diminish acidification. Amines are gaseous bases, whose general formula is RNH2, R2NH or R3N. Due to their effective participation in neutralization it is hard to detect their real atmospheric concentrations. Globally, the main known

945 anthropogenic amine emissions are from animal husbandry, industry and compost processes, and the natural sources of amines are assumed to be ocean, biomass burning, vegetation and soil (Ge et al., 2011). It has been shown that Amines amines also-affect hydroxyl radical (OH) reactivity and therefore all atmospheric chemistry (Hellén et al. 2014, Kieloaho et al. 2013).

- Models based on quantum chemistry data have shown that they-amines could participate in aerosol-new particle formation (NPF) with sulfuric acid even at very low mixing ratios (Kurtén et al. 2008, Paasonen et al. 2012), and also experiments in laboratory have proved formation of aminium salts when amines react with nitric or sulphuric acid (Murphy et al. 2007). Also In addition the recent laboratory experiments at the CLOUD chamber shows that already even at 
[revised manuscript text omitted]

|       | Mean                         | Average          | Average      |
|-------|------------------------------|------------------|--------------|
| Month | <del>Temperature</del>       | Wind speed       | Humidity (%) |
|       | <del>(</del> * <del>C)</del> | <del>(m/s)</del> |              |

| March          | <del>0.4</del>  | <del>2.6</del> | <del>87</del>  |  |
|----------------|-----------------|----------------|----------------|--|
| April          | <del>3.7</del>  | <del>2.4</del> | <del>75</del>  |  |
| May            | <del>9.7</del>  | <del>1.8</del> | <del>69</del>  |  |
| July           | <del>13.8</del> | <del>1.5</del> | <del>75</del>  |  |
| August         | <del>17.8</del> | <del>1.4</del> | 74             |  |
| Novem          | <del>2.7</del>  | <del>2.9</del> | <del>95</del>  |  |
| <del>ber</del> |                 |                |                |  |
| Decem          | -0.1            | <del>1.9</del> | <del>9</del> 4 |  |
| <del>ber</del> |                 |                |                |  |

**1035 2.3 Measurement methods**

[revised manuscript text omitted]

|      | internal standard (ISTD) for all amines. DEA 10 was used, because it behaved same way in IC-separation but had different                                 |   |
|------|---------------------------------------------------------------------------------------------------------------------------------------------------------------------|---|
|      | mass than studied amines. 50.0µl of DEA10 was added to the MARGAs ISTD solution bottle (LiBr). After the ion                                                        |   |
| 1075 | chromatograph the ISTD mixed with the sample entered the MS-detection. DEA 10 was used to correct for possible losses to                                 |   |
|      | instrumentation and correct changes of MS response. A 3-point external calibration was used for all measured alkyl amines                                           | - |
|      | (concentration levels 10, 50 and 300 ng m -3 ). The system was calibrated every two weeks, by stopping the air flow of the                               |   |
|      | MARGA and directing standard solutions to the sample syringe pumps, before analysis by IC-separation and MS-detection.                                              |   |
|      | Ammonia (NH 3 ) and ammonium (NH 4 + ) (the sum of them referred to as NH 8 ) were also measured with MARGA at the same |   |
| 1080 | time with the method described in Makkonen et al. (2012 and 2014), except we used oxalic acid solution for eluent. For                                              |   |
|      | NHg-measurements only conductivity detector was used and the internal standard was lithium bromide (Acros Organics,                                                 |   |
|      | New Jersey, USA). Instrumental blank values for MARGA-MS were measured every month or every other month with                                                        | L |
|      | MARGAs blank-mode: the sample airflow was stopped, and the analysis cycle was running for 6 hours without sampling.                                                 | U |
|      |                                                                                                                                                                     |   |
| 1085 | Deuterated diethyl d 10 amine (Sigma Aldrich: Isotec™; Sigma Aldrich, St. Louis, MO, USA) was used as an internal                                        |   |
|      | et and end for all amines and a 2 maint antennal activation and an and for all ansatzed allest amines (a manufacture la set                                         |   |

Deuterated diethyl-d10-amine (DEA10, Sigma-Aldrich: Isotec™: Sigma-Aldrich, St. Louis, MO, USA) was used as an

[revised manuscript text omitted]

|      | molecular masses and inorganic cations             | (e.g. K+, Mg2+). V                                                                                                                                                                                                                                                                                                                                                                                                                                                                                                                                                                                                                                                                                                                                                                                                                                                                                                                                                                                                                                                                                                                                                                                                                                                                                                                                                                                                                                                                                                                                                                                                                                                                                                                                                                                                                                                                                                                                                                                                                                                                                                                                                                                                                                                                                                                                                                                                                                                                                                                                                                                                                                                                                                                                                                                                                                                                                                                                                                                                      | /erriele et al. (2012          | 2) developed also an 1  | IC-MS method for amines        |   |                                            |
|------|----------------------------------------------------|-------------------------------------------------------------------------------------------------------------------------------------------------------------------------------------------------------------------------------------------------------------------------------------------------------------------------------------------------------------------------------------------------------------------------------------------------------------------------------------------------------------------------------------------------------------------------------------------------------------------------------------------------------------------------------------------------------------------------------------------------------------------------------------------------------------------------------------------------------------------------------------------------------------------------------------------------------------------------------------------------------------------------------------------------------------------------------------------------------------------------------------------------------------------------------------------------------------------------------------------------------------------------------------------------------------------------------------------------------------------------------------------------------------------------------------------------------------------------------------------------------------------------------------------------------------------------------------------------------------------------------------------------------------------------------------------------------------------------------------------------------------------------------------------------------------------------------------------------------------------------------------------------------------------------------------------------------------------------------------------------------------------------------------------------------------------------------------------------------------------------------------------------------------------------------------------------------------------------------------------------------------------------------------------------------------------------------------------------------------------------------------------------------------------------------------------------------------------------------------------------------------------------------------------------------------------------------------------------------------------------------------------------------------------------------------------------------------------------------------------------------------------------------------------------------------------------------------------------------------------------------------------------------------------------------------------------------------------------------------------------------------------------|--------------------------------|-------------------------|--------------------------------|---|--------------------------------------------|
|      | with offline sampling with midget impi             | ngers. They also no                                                                                                                                                                                                                                                                                                                                                                                                                                                                                                                                                                                                                                                                                                                                                                                                                                                                                                                                                                                                                                                                                                                                                                                                                                                                                                                                                                                                                                                                                                                                                                                                                                                                                                                                                                                                                                                                                                                                                                                                                                                                                                                                                                                                                                                                                                                                                                                                                                                                                                                                                                                                                                                                                                                                                                                                                                                                                                                                                                                                     | ticed that adding N            | AS detection after a co | onductivity detector           |   |                                            |
|      | overcomes the co-eluting problem of IC             | separation. They h                                                                                                                                                                                                                                                                                                                                                                                                                                                                                                                                                                                                                                                                                                                                                                                                                                                                                                                                                                                                                                                                                                                                                                                                                                                                                                                                                                                                                                                                                                                                                                                                                                                                                                                                                                                                                                                                                                                                                                                                                                                                                                                                                                                                                                                                                                                                                                                                                                                                                                                                                                                                                                                                                                                                                                                                                                                                                                                                                                                                      | ad a 4-step gradie             | nt elution in their met | thod, and suppression          |   |                                            |
| 1140 | before the conductivity detector. We wa            | inted to keep our m                                                                                                                                                                                                                                                                                                                                                                                                                                                                                                                                                                                                                                                                                                                                                                                                                                                                                                                                                                                                                                                                                                                                                                                                                                                                                                                                                                                                                                                                                                                                                                                                                                                                                                                                                                                                                                                                                                                                                                                                                                                                                                                                                                                                                                                                                                                                                                                                                                                                                                                                                                                                                                                                                                                                                                                                                                                                                                                                                                                                     | ethod as simple as             | possible to make it e   | asy to use in the field, and   |   |                                            |
|      | isocratic elution without suppression wa           | as good in that purp                                                                                                                                                                                                                                                                                                                                                                                                                                                                                                                                                                                                                                                                                                                                                                                                                                                                                                                                                                                                                                                                                                                                                                                                                                                                                                                                                                                                                                                                                                                                                                                                                                                                                                                                                                                                                                                                                                                                                                                                                                                                                                                                                                                                                                                                                                                                                                                                                                                                                                                                                                                                                                                                                                                                                                                                                                                                                                                                                                                                    | ose.                           | -                       |                                |   |                                            |
|      | The sub-la exclusion are denoted in th             | - 6-14 4b4b                                                                                                                                                                                                                                                                                                                                                                                                                                                                                                                                                                                                                                                                                                                                                                                                                                                                                                                                                                                                                                                                                                                                                                                                                                                                                                                                                                                                                                                                                                                                                                                                                                                                                                                                                                                                                                                                                                                                                                                                                                                                                                                                                                                                                                                                                                                                                                                                                                                                                                                                                                                                                                                                                                                                                                                                                                                                                                                                                                                                             | - 4 h - 4 1 ( 6     |                         |                                | C |                                            |
|      | The whole analysis was conducted in the     | 1 DA 1:11                                                                                                                                                                                                                                                                                                                                                                                                                                                                                                                                                                                                                                                                                                                                                                                                                                                                                                                                                                                                                                                                                                                                                                                                                                                                                                                                                                                                                                                                                                                                                                                                                                                                                                                                                                                                                                                                                                                                                                                                                                                                                                                                                                                                                                                                                                                                                                                                                                                                                                                                                                                                                                                                                                                                                                                                                                                                                                                                                                                                               |                                | rom sample transporta   | ation. However, the            |   | Formatted: Normal, Line spacing: 1.5 lines |
|      | drawback in the analysis was that DEA              | and BA, which hav                                                                                                                                                                                                                                                                                                                                                                                                                                                                                                                                                                                                                                                                                                                                                                                                                                                                                                                                                                                                                                                                                                                                                                                                                                                                                                                                                                                                                                                                                                                                                                                                                                                                                                                                                                                                                                                                                                                                                                                                                                                                                                                                                                                                                                                                                                                                                                                                                                                                                                                                                                                                                                                                                                                                                                                                                                                                                                                                                                                                       | e the same molect              | llar masses, did not se | eparate completely. From a     |   |                                            |
|      | technical point of view one of the drawl           | backs of the MARG                                                                                                                                                                                                                                                                                                                                                                                                                                                                                                                                                                                                                                                                                                                                                                                                                                                                                                                                                                                                                                                                                                                                                                                                                                                                                                                                                                                                                                                                                                                                                                                                                                                                                                                                                                                                                                                                                                                                                                                                                                                                                                                                                                                                                                                                                                                                                                                                                                                                                                                                                                                                                                                                                                                                                                                                                                                                                                                                                                                                       | A-MS was that th               | e system was quite vi   | alnerable. We lost many |   |                                            |
| 1145 | measuring days because some part of th             | e system was broke                                                                                                                                                                                                                                                                                                                                                                                                                                                                                                                                                                                                                                                                                                                                                                                                                                                                                                                                                                                                                                                                                                                                                                                                                                                                                                                                                                                                                                                                                                                                                                                                                                                                                                                                                                                                                                                                                                                                                                                                                                                                                                                                                                                                                                                                                                                                                                                                                                                                                                                                                                                                                                                                                                                                                                                                                                                                                                                                                                                                      | en. The MARGA s                | ide also needed ~401    | solutions (e.g. eluents,       |   |                                            |
|      | absorbation solution for sampling, and i           | nternal standard sol                                                                                                                                                                                                                                                                                                                                                                                                                                                                                                                                                                                                                                                                                                                                                                                                                                                                                                                                                                                                                                                                                                                                                                                                                                                                                                                                                                                                                                                                                                                                                                                                                                                                                                                                                                                                                                                                                                                                                                                                                                                                                                                                                                                                                                                                                                                                                                                                                                                                                                                                                                                                                                                                                                                                                                                                                                                                                                                                                                                                    | lution) that needed            | to be changed weekl     | y. The ESI-chamber of the      |   |                                            |
|      | MS needed to be cleaned weekly, becau              | se oxalic acid was                                                                                                                                                                                                                                                                                                                                                                                                                                                                                                                                                                                                                                                                                                                                                                                                                                                                                                                                                                                                                                                                                                                                                                                                                                                                                                                                                                                                                                                                                                                                                                                                                                                                                                                                                                                                                                                                                                                                                                                                                                                                                                                                                                                                                                                                                                                                                                                                                                                                                                                                                                                                                                                                                                                                                                                                                                                                                                                                                                                                      | crystallizing into in          | t. Despite the drawbac  | cks, to our knowledge with     |   |                                            |
|      | the MARGA-MS method we achieved                    | he largest data set o                                                                                                                                                                                                                                                                                                                                                                                                                                                                                                                                                                                                                                                                                                                                                                                                                                                                                                                                                                                                                                                                                                                                                                                                                                                                                                                                                                                                                                                                                                                                                                                                                                                                                                                                                                                                                                                                                                                                                                                                                                                                                                                                                                                                                                                                                                                                                                                                                                                                                                                                                                                                                                                                                                                                                                                                                                                                                                                                                                                                   | of amine concentra             | tions available at the  | moment.                        | ( | Formatted: English (U.S.)                  |
|      |                                                    |                                                                                                                                                                                                                                                                                                                                                                                                                                                                                                                                                                                                                                                                                                                                                                                                                                                                                                                                                                                                                                                                                                                                                                                                                                                                                                                                                                                                                                                                                                                                                                                                                                                                                                                                                                                                                                                                                                                                                                                                                                                                                                                                                                                                                                                                                                                                                                                                                                                                                                                                                                                                                                                                                                                                                                                                                                                                                                                                                                                                                         |                                |                         |                                |   |                                            |
|      |                                                    |                                                                                                                                                                                                                                                                                                                                                                                                                                                                                                                                                                                                                                                                                                                                                                                                                                                                                                                                                                                                                                                                                                                                                                                                                                                                                                                                                                                                                                                                                                                                                                                                                                                                                                                                                                                                                                                                                                                                                                                                                                                                                                                                                                                                                                                                                                                                                                                                                                                                                                                                                                                                                                                                                                                                                                                                                                                                                                                                                                                                                         |                                |                         |                                |   |                                            |
| 1150 |                                                    |                                                                                                                                                                                                                                                                                                                                                                                                                                                                                                                                                                                                                                                                                                                                                                                                                                                                                                                                                                                                                                                                                                                                                                                                                                                                                                                                                                                                                                                                                                                                                                                                                                                                                                                                                                                                                                                                                                                                                                                                                                                                                                                                                                                                                                                                                                                                                                                                                                                                                                                                                                                                                                                                                                                                                                                                                                                                                                                                                                                                                         |                                |                         |                                |   |                                            |
| 1100 |                                                    |                                                                                                                                                                                                                                                                                                                                                                                                                                                                                                                                                                                                                                                                                                                                                                                                                                                                                                                                                                                                                                                                                                                                                                                                                                                                                                                                                                                                                                                                                                                                                                                                                                                                                                                                                                                                                                                                                                                                                                                                                                                                                                                                                                                                                                                                                                                                                                                                                                                                                                                                                                                                                                                                                                                                                                                                                                                                                                                                                                                                                         |                                |                         |                                |   |                                            |
|      |                                                    |                                                                                                                                                                                                                                                                                                                                                                                                                                                                                                                                                                                                                                                                                                                                                                                                                                                                                                                                                                                                                                                                                                                                                                                                                                                                                                                                                                                                                                                                                                                                                                                                                                                                                                                                                                                                                                                                                                                                                                                                                                                                                                                                                                                                                                                                                                                                                                                                                                                                                                                                                                                                                                                                                                                                                                                                                                                                                                                                                                                                                         |                                |                         |                                |   |                                            |
|      |                                                    |                                                                                                                                                                                                                                                                                                                                                                                                                                                                                                                                                                                                                                                                                                                                                                                                                                                                                                                                                                                                                                                                                                                                                                                                                                                                                                                                                                                                                                                                                                                                                                                                                                                                                                                                                                                                                                                                                                                                                                                                                                                                                                                                                                                                                                                                                                                                                                                                                                                                                                                                                                                                                                                                                                                                                                                                                                                                                                                                                                                                                         |                                |                         |                                |   |                                            |
|      |                                                    |                                                                                                                                                                                                                                                                                                                                                                                                                                                                                                                                                                                                                                                                                                                                                                                                                                                                                                                                                                                                                                                                                                                                                                                                                                                                                                                                                                                                                                                                                                                                                                                                                                                                                                                                                                                                                                                                                                                                                                                                                                                                                                                                                                                                                                                                                                                                                                                                                                                                                                                                                                                                                                                                                                                                                                                                                                                                                                                                                                                                                         |                                |                         |                                |   |                                            |
|      |                                                    |                                                                                                                                                                                                                                                                                                                                                                                                                                                                                                                                                                                                                                                                                                                                                                                                                                                                                                                                                                                                                                                                                                                                                                                                                                                                                                                                                                                                                                                                                                                                                                                                                                                                                                                                                                                                                                                                                                                                                                                                                                                                                                                                                                                                                                                                                                                                                                                                                                                                                                                                                                                                                                                                                                                                                                                                                                                                                                                                                                                                                         |                                |                         |                                |   |                                            |
|      |                                                    |                                                                                                                                                                                                                                                                                                                                                                                                                                                                                                                                                                                                                                                                                                                                                                                                                                                                                                                                                                                                                                                                                                                                                                                                                                                                                                                                                                                                                                                                                                                                                                                                                                                                                                                                                                                                                                                                                                                                                                                                                                                                                                                                                                                                                                                                                                                                                                                                                                                                                                                                                                                                                                                                                                                                                                                                                                                                                                                                                                                                                         |                                |                         |                                |   |                                            |
|      |                                                    |                                                                                                                                                                                                                                                                                                                                                                                                                                                                                                                                                                                                                                                                                                                                                                                                                                                                                                                                                                                                                                                                                                                                                                                                                                                                                                                                                                                                                                                                                                                                                                                                                                                                                                                                                                                                                                                                                                                                                                                                                                                                                                                                                                                                                                                                                                                                                                                                                                                                                                                                                                                                                                                                                                                                                                                                                                                                                                                                                                                                                         |                                |                         |                                |   |                                            |
|      |                                                    |                                                                                                                                                                                                                                                                                                                                                                                                                                                                                                                                                                                                                                                                                                                                                                                                                                                                                                                                                                                                                                                                                                                                                                                                                                                                                                                                                                                                                                                                                                                                                                                                                                                                                                                                                                                                                                                                                                                                                                                                                                                                                                                                                                                                                                                                                                                                                                                                                                                                                                                                                                                                                                                                                                                                                                                                                                                                                                                                                                                                                         |                                |                         |                                |   |                                            |
| 1155 |                                                    |                                                                                                                                                                                                                                                                                                                                                                                                                                                                                                                                                                                                                                                                                                                                                                                                                                                                                                                                                                                                                                                                                                                                                                                                                                                                                                                                                                                                                                                                                                                                                                                                                                                                                                                                                                                                                                                                                                                                                                                                                                                                                                                                                                                                                                                                                                                                                                                                                                                                                                                                                                                                                                                                                                                                                                                                                                                                                                                                                                                                                         |                                |                         |                                |   |                                            |
|      | Table 1 Detection limits (DL) of differ            |                                                                                                                                                                                                                                                                                                                                                                                                                                                                                                                                                                                                                                                                                                                                                                                                                                                                                                                                                                                                                                                                                                                                                                                                                                                                                                                                                                                                                                                                                                                                                                                                                                                                                                                                                                                                                                                                                                                                                                                                                                                                                                                                                                                                                                                                                                                                                                                                                                                                                                                                                                                                                                                                                                                                                                                                                                                                                                                                                                                                                         | :tt                            | C                |                                |   |                                            |
|      | made using conversion factor $ppt_{n} = c(r)$      | $(0.0409 \times (10.0409 \times (10.0409)))))))))$ )))) | ( W )) by Finlaysor     | -Pitts (2000), with M   | IW the molar mass of the       |   |                                            |
|      | amine, ammonia or ammonium The pi                  | ecision for IC-MS                                                                                                                                                                                                                                                                                                                                                                                                                                                                                                                                                                                                                                                                                                                                                                                                                                                                                                                                                                                                                                                                                                                                                                                                                                                                                                                                                                                                                                                                                                                                                                                                                                                                                                                                                                                                                                                                                                                                                                                                                                                                                                                                                                                                                                                                                                                                                                                                                                                                                                                                                                                                                                                                                                                                                                                                                                                                                                                                                                                                       | analysis was defin             | ed by calculating stan  | ndard deviations of liquid     |   |                                            |
|      | 200 ng m -3 standard measured 6 times i | n a row. In the data                                                                                                                                                                                                                                                                                                                                                                                                                                                                                                                                                                                                                                                                                                                                                                                                                                                                                                                                                                                                                                                                                                                                                                                                                                                                                                                                                                                                                                                                                                                                                                                                                                                                                                                                                                                                                                                                                                                                                                                                                                                                                                                                                                                                                                                                                                                                                                                                                                                                                                                                                                                                                                                                                                                                                                                                                                                                                                                                                                                                    | series there were l            | both gas and particle s | side measurements. The         |   |                                            |
| 1160 | accuracy for IC-MS analysis was calcul             | ated by subtracting                                                                                                                                                                                                                                                                                                                                                                                                                                                                                                                                                                                                                                                                                                                                                                                                                                                                                                                                                                                                                                                                                                                                                                                                                                                                                                                                                                                                                                                                                                                                                                                                                                                                                                                                                                                                                                                                                                                                                                                                                                                                                                                                                                                                                                                                                                                                                                                                                                                                                                                                                                                                                                                                                                                                                                                                                                                                                                                                                                                                     | the averages of th             | e data series describe  | d earlier from the expected    |   |                                            |
|      | values, dividing those with the expected           | I values and multipl                                                                                                                                                                                                                                                                                                                                                                                                                                                                                                                                                                                                                                                                                                                                                                                                                                                                                                                                                                                                                                                                                                                                                                                                                                                                                                                                                                                                                                                                                                                                                                                                                                                                                                                                                                                                                                                                                                                                                                                                                                                                                                                                                                                                                                                                                                                                                                                                                                                                                                                                                                                                                                                                                                                                                                                                                                                                                                                                                                                                    | lying them by 100 4 | %.               |                                |   |                                            |
|      | Amine                                              | DL (ng m -3 )                                                                                                                                                                                                                                                                                                                                                                                                                                                                                                                                                                                                                                                                                                                                                                                                                                                                                                                                                                                                                                                                                                                                                                                                                                                                                                                                                                                                                                                                                                                                                                                                                                                                                                                                                                                                                                                                                                                                                                                                                                                                                                                                                                                                                                                                                                                                                                                                                                                                                                                                                                                                                                                                                                                                                                                                                                                                                                                                                                                                | DL (ppt v )         | Precision (%)           | Accuracy (%)                   |   | Formatted: Line spacing: single            |
|      | MMA both gas and aerosol                           | 2.4                                                                                                                                                                                                                                                                                                                                                                                                                                                                                                                                                                                                                                                                                                                                                                                                                                                                                                                                                                                                                                                                                                                                                                                                                                                                                                                                                                                                                                                                                                                                                                                                                                                                                                                                                                                                                                                                                                                                                                                                                                                                                                                                                                                                                                                                                                                                                                                                                                                                                                                                                                                                                                                                                                                                                                                                                                                                                                                                                                                                                     | 19                             | 10                      | 24                             |   | Formatted Table                            |
|      |                                                    |                                                                                                                                                                                                                                                                                                                                                                                                                                                                                                                                                                                                                                                                                                                                                                                                                                                                                                                                                                                                                                                                                                                                                                                                                                                                                                                                                                                                                                                                                                                                                                                                                                                                                                                                                                                                                                                                                                                                                                                                                                                                                                                                                                                                                                                                                                                                                                                                                                                                                                                                                                                                                                                                                                                                                                                                                                                                                                                                                                                                                  | 1.2                            | 10                      |                         |   | Formatted: Line spacing: single            |
|      | DMA, (March to August) gas                         | 3.1                                                                                                                                                                                                                                                                                                                                                                                                                                                                                                                                                                                                                                                                                                                                                                                                                                                                                                                                                                                                                                                                                                                                                                                                                                                                                                                                                                                                                                                                                                                                                                                                                                                                                                                                                                                                                                                                                                                                                                                                                                                                                                                                                                                                                                                                                                                                                                                                                                                                                                                                                                                                                                                                                                                                                                                                                                                                                                                                                                                                              | 1.7                     | 11               | 31 •                    | ( | Formatted: Line spacing: single            |
|      | aerosols                                           |                                                                                                                                                                                                                                                                                                                                                                                                                                                                                                                                                                                                                                                                                                                                                                                                                                                                                                                                                                                                                                                                                                                                                                                                                                                                                                                                                                                                                                                                                                                                                                                                                                                                                                                                                                                                                                                                                                                                                                                                                                                                                                                                                                                                                                                                                                                                                                                                                                                                                                                                                                                                                                                                                                                                                                                                                                                                                                                                                                                                                         |                                |                         |                                |   |                                            |

| (November to December) gas        | 1.1  |             |           |           |                                        |
|-----------------------------------|-------------|-------------|-----------|-----------|----------------------------------------|
| aerosols                          | 0.37 | 0.20 |           |           |                                        |
|                                   | 0.76 |

---

## Author Response (AR2)

**The Authors have made significant revisions to their manuscript and greatly strengthened its contribution to our understanding of atmospheric amines. In particular, the added information regarding the use of internal standards significantly improves confidence in the quantitative capabilities of MARGA-MS despite the frequently lost measurements due to instrumental technical issues. There are only a few**
5 **technical changes that should be made the manuscript prior to publication in ACP.**

We thank the Referee for reading the manuscript again and giving good and helpful comments.

**Lines 38-39: Ammonia is converted to nitrate by soil microbes, giving it an overall acidifying effect on ecosystems when deposited. I am not aware of any work demonstrating that amines do not also have a net acidifying effect. Recommend addition of a reference here or removing this sentence.**

10 We took the sentence of.

**Lines 152-153: The measurements are well below the calibration range for the alkyl amines and the authors have not been transparent about this in their discussion. It is easily appreciated that the calibrations were performed in the field at expected levels. Addition of this fact in the discussion would improve the scientific integrity of this work.**

15 We added to row 153 "in the field" in the end of the sentence, and we added to the end of first chapter in section 3.1 that

"Calibration levels (10, 50 and 300 ng m$^{-3}$) were selected so, that the lowest level was a bit higher than biggest alkyl amine detection limit (DMA, 3.1 ng m$^{-3}$), and highest was too high to exceed in the measurements. When the measurements started, most of the data was under the lowest calibration point, which increased the
20 uncertainty."

**Lines 158-159: It is more clear to state 'were measured at least every other month with MARGAs blank-mode'**

We have changed that.

**Lines 213-214: Remove the last sentence here. Claiming the most total measurements undermines the value of this body of work where it explores relationships between the amines and environmental parameters, which is much more impactful. These points are a consequence of the large number of measurements and far more important to highlight as a scientific contribution.**

30 The sentence was removed.

**Line 230: Move this entire section and Figure 1 to the SI**

We did, and we added a sentence in the end of Chapter 3.1 to reference to the section.

35 **Line 270/Figure 2: The y-axis labels have not been typeset properly and neither have some of the labels. Recommend careful review of all table and figure formatting as there are a number of similar issues throughout the manuscript.**

We changed the y-axis and checked all tables and figures.

40 **Line 504/Figure 10: This is not referenced in the manuscript and should be, or else it should be moved to the SI or removed altogether.**

We have added more noticeable reference.

[revised manuscript text omitted]